# Continuous Microfluidic Antisolvent Crystallization as a Bottom-Up Solution for the Development of Long-Acting Injectable Formulations

**DOI:** 10.3390/pharmaceutics16030376

**Published:** 2024-03-08

**Authors:** Snehashis Nandi, Laura Verstrepen, Mariana Hugo Silva, Luis Padrela, Lidia Tajber, Alain Collas

**Affiliations:** 1Janssen Pharmaceutica NV, Johnson & Johnson Innovative Medicine, 2340 Beerse, Belgium; snehashis.nandi@ul.ie (S.N.); lverstr2@its.jnj.com (L.V.); mhbbsilva@gmail.com (M.H.S.); 2SSPC—The Science Foundation Ireland Research Centre for Pharmaceuticals, V94 T9PX Limerick, Ireland; luis.padrela@ul.ie (L.P.); ltajber@tcd.ie (L.T.); 3Department of Chemical Sciences, Bernal Institute, University of Limerick, V94 T9PX Limerick, Ireland; 4Faculty of Pharmaceutical Sciences, University of Antwerp, 2000 Antwerp, Belgium; 5School of Pharmacy and Pharmaceutical Sciences, Trinity College Dublin, College Green, D02 PN40 Dublin, Ireland

**Keywords:** itraconazole, LAI, SCT-CLASC, PSD, Vit E TPGS 1000, microsuspension

## Abstract

A bottom-up approach was investigated to produce long-acting injectable (LAI) suspension-based formulations to overcome specific limitations of top-down manufacturing methods by tailoring drug characteristics while making the methods more sustainable and cost-efficient. A Secoya microfluidic crystallization technology-based continuous liquid antisolvent crystallization (SCT-CLASC) process was optimized and afterward compared to an earlier developed microchannel reactor-based continuous liquid antisolvent crystallization (MCR-CLASC) setup, using itraconazole (ITZ) as the model drug. After operating parameter optimization and downstream processing (i.e., concentrating the suspensions), stable microsuspensions were generated with a final solid loading of 300 mg ITZ/g suspension. The optimized post-precipitation feed suspension consisted of 40 mg ITZ/g suspension with a drug-to-excipient ratio of 53:1. Compared to the MCR-CLASC setup, where the post-precipitation feed suspensions contained 10 mg ITZ/g suspension and had a drug-to-excipient ratio of 2:1, a higher drug concentration and lower excipient use were successfully achieved to produce LAI microsuspensions using the SCT-CLASC setup. To ensure stability during drug crystallization and storage, the suspensions’ quality was monitored for particle size distribution (PSD), solid-state form, and particle morphology. The PSD of the ITZ crystals in suspension was maintained within the target range of 1–10 µm, while the crystals displayed an elongated plate-shaped morphology and the solid state was confirmed to be form I, which is the most thermodynamically stable form of ITZ. In conclusion, this work lays the foundation for the SCT-CLASC process as an energy-efficient, robust, and reproducible bottom-up approach for the manufacture of LAI microsuspensions using ITZ at an industrial scale.

## 1. Introduction

Long-acting injectable (LAI) formulations are parenteral delivery systems that provide a sustained release and thereby a steady exposure to APIs intended to be delivered for prolonged periods that range from days to months, in contrast to oral formulations [1,2,3,4]. LAI formulations improve quality of life and patient compliance by substantially minimizing administration frequency as well as undesired side effects [1,4,5]. For this reason, LAIs can provide huge benefits for patients suffering from dysphagia or chronic diseases, such as mental disorders, HIV infection, tuberculosis, and hormone replacement treatments, where limited patient adherence may have a negative outcome on treatment efficacy [6,7,8,9]. LAIs are primarily delivered through the intramuscular (IM) or subcutaneous (SC) routes as oil solutions, in situ forming gels, microspheres, implants, or micro- or nanosuspensions [3,10,11]. Owing to concerns over side effects, patient tolerance, and pain upon injection in most of the delivery systems, the micro- and nanosuspension technology has significantly advanced, and many of the more recent approved LAIs are either nano- or microsuspensions [4,12,13]. They are typically composed of poorly water-soluble BCS class II or IV APIs as micro- or nanocrystals, which are usually stabilized with one or more polymers and/or surfactants [14]. All current marketed suspension-based LAI formulations are manufactured using top-down methods. Top-down manufacturing approaches involve physical processes of mechanical attrition to reduce the particle sizes of APIs to micro- or nanoparticle scales. Conventional methods for size reduction in larger drug crystals include wet media milling and high-pressure homogenization [15,16,17]. The main challenges of these approaches are high energy requirements and large mechanical stresses exerted on the APIs, potentially resulting in inconsistent crystal solid forms in different batches and product contamination [7,18,19]. To reduce the limitations mentioned, bottom-up approaches (building particles from the molecular level) are emerging as alternatives to producing LAI suspensions [5,15]. Liquid antisolvent crystallization (LASC) has been gaining significance as a potential bottom-up process for producing aqueous suspension-based LAIs, since, inherently, it should allow for better control over the surface and shape properties of API particles with respect to top-down methods [20].

Hugo Silva et al. pointed out that the LASC process can indeed be used as an alternative energy-efficient bottom-up method for the production of aqueous suspension-based LAIs with a reduced risk of contamination from milling equipment and fewer processing steps and that it may prove to be comparable in terms of stability and particle size distribution (PSD) to current industrially accepted top-down approaches [21,22,23]. Others, too, have demonstrated that this approach allows finetuning and control of drug product characteristics, such as PSD, crystallinity, and morphology, and that it is a simple, more sustainable, and less expensive process [24,25,26]. Thorat et al. pointed out that studies in the literature reporting evidence of the utility of the LASC process for the generation of nano- and microsuspension formulations cover over 50 APIs [27]. In the case of a continuous LASC process, the drug solution and antisolvent are injected together into a mixing platform, such as a microchannel reactor [28], a tubular baffled crystallizer [29,30], or a microfluidic device [31]. The continuous approach is straightforward, quick to scale up, requires fewer steps in the process, and uses low-cost conventional equipment, and is, therefore, relevant and appealing to pharmaceutical manufacturing [25]. Nevertheless, the application of continuous crystallization remains challenging because of existing drawbacks, such as fouling and a propensity for clogging [32] and encrustation [32,33] (i.e., nucleation and growth of particles on the reactor wall).

Microfluidics technology employing microchannels along with microscale mixers is evaluated in detail for LASC processes to produce scalable micro- and nanoparticles due to its high nucleation rate [31] and subsequent prompt cease of crystal growth [34,35]. In microfluidic processes, supersaturation is rapidly generated as a result of antisolvents, and solvent addition and intense mixing are achieved through a static micromixer [36,37]. The reaction/diffusion zone is dispersed along the channel downstream because of the purely diffusive mixing [38,39,40]. This improves the interface micromixing efficiency and regulates supersaturation with a highly precise spatial and temporal distribution in a high-efficiency micromass transfer process [28,30,41,42,43]. Since the residence time and the physical and chemical environment are consistent, the particles are produced by the microfluidic process with controlled PSD, morphology, and polymorphism [44]. The geometry of the microchannel can also enhance mixing and thus affect the final characteristics of the particles produced, such as morphology [15,24,45]. Static mixers were used to promote homogeneous mixing of an active pharmaceutical ingredient (API) solution and an antisolvent to obtain smaller crystals with a narrower size distribution [46,47]. Liu et al. performed continuous antisolvent crystallization of dolutegravir sodium using a microfluidic device for the production of small particles (D50 values of 5–10 μm) with a narrow PSD (the range of the width at 4–7 μm) [48]. Furthermore, according to the literature, the microfluidic approach offers control over how and where material nucleates, eliminating undesirable nucleation on reactor walls, which, if left uncontrolled, leads to fouling, back-mixing, and eventually obstruction. Flow-focusing geometries can guide nucleation away from reactor walls, reducing encrustation in microreactors even further [34,49,50].

The simple but efficient Secoya microfluidic crystallization technology (SCT-LAB) of Secoya Technologies [51] provides evidence of real-time monitoring facility and better control of process parameters (i.e., temperature, solvent–antisolvent stream flow rate, and solvent-to-antisolvent ratio) during the entire operational window. This further allows improvement of the reproducibility and robustness of the process when compared to conventional batch systems [52]. This microfluidic technology with high manufacturing capacity and low fabrication costs is an interesting alternative to conventional batch systems [40,53].

Before adopting the microfluidic-based LASC process for industrial production of LAI suspensions, certain hurdles must be overcome, such as the lack of process development for large-scale production and adaptable industrial manufacturing equipment and the need for integration in the commercial setup to create a standardized, automated, and robust high-throughput process [17,22], along with formulation processing variable optimization, such as selection of the right excipient type, minimization of the excipient concentration, and boosting of the API solid loading [4,17,19].

In the present work, a continuous microfluidic liquid antisolvent crystallization (SCT-CLASC) process was developed using itraconazole (ITZ)–water–NMP as a model system and a Secoya SCT-LAB microfluidic device. The methodology was utilized to produce stable LAI suspensions [19] with PSDs between 1 and 10 µm, with higher API loading and/or lower excipient use compared to the pre-optimized continuous microchannel reactor-based LASC process (MCR-CLASC) described in an earlier paper [54]. This work investigates the impact of the solvent-to-antisolvent ratio, the solvent-to-antisolvent stream flow rate, the type of static mixing device, and the effect of the stabilizer type and concentration on the post-precipitation feed suspension using a one-factor-at-a-time (OFAT) approach. To ensure stability during manufacturing, storage, and administration, the quality of the produced suspensions was monitored in terms of PSD, solid-state form, and morphology. The impact of downstream processing on the stability and quality of the generated LAI suspensions was investigated. A critical comparison in terms of in vitro drug release from final LAI suspensions is presented.

## 2. Experimental Section

### 2.1. Materials

Itraconazole (ITZ) (CAS no. 84625-61-6) with >99% purity was supplied by Janssen Pharmaceutica NV (Beerse, Belgium) and was used as a model API for the continuous LASC process. N-methyl-2-pyrrolidone (NMP) (CAS no. 872-50-4), HPLC grade of purity >99.5% and Ph. Eur. grade >99%, were purchased from Sigma-Aldrich (Steinheim, Germany) and Fisher Scientific (Loughborough, UK). D-α-tocopherol polyethylene glycol 1000 succinate (Vit E TPGS 1000) of Ph. Eur. grade and USP grade was purchased from Sigma–Aldrich Chemie GmbH (Steinheim, Germany) and from BASF Chemtrade GmbH (Burghernheim, Germany). Sodium carboxymethyl cellulose (NaCMC), poloxamer (POL) 188, POL 338, POL 407, and polyvinylpyrrolidone (PVP) K30 were purchased from Sigma Life Science (Cork, Ireland).

Type I ultrapure deionized water was acquired from a Milli-Q^®^ integral water purification system (Milli-Q Advantage A10; Merck Millipore, Hellerup, Denmark). Scientific Commodities Inc. (Lake Havasu City, AZ, USA), GL Sciences (Rolling Hills Estates, CA, USA), and IDEX Health & Science LLC (West Henrietta, NY, USA), provided the perfluoroalkoxy (PFA) tubing (0.33 mm inner diameter) for the inlet, the fluorinated ethylene propylene (FEP) tubing (0.5 mm inner diameter) for the outlet, and the polyether ether ketone (PEEK) microchannel reactor (Y mixer) with a 0.5 mm through hole. HPLC-grade acetonitrile was also obtained from Merck KGaA (Darmstadt, Germany). Compressed nitrogen, obtained from Messer Belgium NV (Zwijndrecht, Belgium), was used to pressurize the membrane setups. Different membranes were used for the filtration strategies. The centrifugal filter devices and a Microsep^TM^ Advance device (0.45 µm) were obtained from the Pall Corporation (Fajardo, Puerto Rico). Durapore^TM^ Membrane Filters (0.45 µm) were purchased from Merck Millipore (Carrigtwohill, Ireland). No pre-treatments were applied to the membranes. All solutions and suspensions were filtered with a Durapore^®^ 0.22 µm PVDF membrane (Merck Millipore Ltd., Cork, Ireland) before analysis by PXRD.

### 2.2. MCR-CLASC Process

An optimized MCR-CLASC setup was assembled to generate ITZ suspensions in continuous mode, as explained in detail in the earlier work published by our group [54]. Two sets of high-precision Teledyne ISCO pumps were connected through a PEEK microchannel reactor (using a Y mixer as a fluidic mixing device) with an internal diameter of 0.5 mm via PFA tubing with a diameter of 0.33 mm and a residence volume of 1.7 µL. The MCR-CLASC process was carried out using ITZ in NMP (10 g in 100 g solution) as solvent and 0.5% *w*/*w* Vit E TPGS 1000 aqueous solution as antisolvent. After crystallization, the suspension was collected in a collection vessel and stirring was maintained at 25 °C for 48 h, as described in the earlier work. The stirring rate (800 rpm), ITZ feed solution concentration (100 mg/g), experimental temperature (25 °C), and aging time before PSD measurements were kept unchanged while varying the S:AS ratio. The solvent and antisolvent flow rates were set at 50 and 200 mL/min with an S:AS ratio of 1:4, 50 and 250 mL/min with an S:AS ratio of 1:5, and 50 and 400 mL/min with an S:AS ratio of 1:8.

### 2.3. Secoya Microfluidic Crystallization Technology-Based Continuous Liquid Antisolvent Crystallization (SCT-CLASC) Process

The tested microfluidic-based continuous crystallization system was SCT-LAB, obtained from Secoya Technologies (Louvain-la-Neuve, Belgium). In the SCT-LAB system, there were two sets of high-precision syringe pumps of 20 mL volume, one for the solvent line and one for the antisolvent line, that were connected through a microfluidic static mixing device (e.g., a co-flow mixer, a T-cross mixer, or a Y mixer) obtained from Secoya Technologies and connected to a 5 m long tubular reactor with a 0.5 mm inner diameter kept under thermostatic conditions, followed by a collection vessel. The whole setup was under a temperature control system, and the temperature was set at 25 °C. The generated suspensions were collected via an FEP outlet tube in an agitated crystallizer at 25 °C and kept under constant stirring at 800 rpm for 48 h before PSD analysis.

The schematic setup of the system is shown in Figure 1. Both syringes’ operational flow rates were 1–40 mL/min for the solvent line and 1–50 mL/min for the antisolvent line. Both continuous phases were filtered through a Durapore^®^ 0.22 µm PVDF membrane (Merck Millipore Ltd., Cork, Ireland) prior to the SCT-CLASC process. During the experiment, the solvent and antisolvent were simultaneously pumped into the mixing device at a fixed flow rate. After every run, all lines were cleaned with acetone and dried with air. When the microfluidic device was blocked, due to early crystallization, it was cleaned using acetic acid and a vacuum. 

### 2.4. Operating Parameter Optimization

#### 2.4.1. Solvent-to-Antisolvent (S:AS) Ratio 

The influence of the solvent-to-antisolvent volumetric (S:AS) ratio on the PSDs and solid states of the generated ITZ suspensions was determined using both co-flow and T-cross mixers. With the co-flow mixer, the S:AS ratios were 1:3, 1:4, 1:5, 1:6, and 1:8 and were varied by increasing the antisolvent proportions, while, with the T-cross mixer, the S:AS ratios evaluated were 1:4, 1:5, and 1:8. The S–AS stream flow rates were changed simultaneously from 10–30 mL/min to 5–40 mL/min according to the S:AS ratios. All other operating parameters were kept constant, with 100 mg/g ITZ in NMP as the feed solution concentration, 0.5% *w*/*w* Vit E TPGS 1000 aqueous solution as the antisolvent, an experimental and storage temperature of 25 °C, a 40 mL volume of final suspension collected, and an aging time of 48 h in the agitated crystallizer before PSD and solid-state analysis.

#### 2.4.2. Solvent–Antisolvent (S–AS) Stream Flow Rate

The influence of the solvent and antisolvent volumetric (S–AS) stream flow rate on the final suspension quality was investigated using a co-flow mixer and changed from 1–4 mL/min (lowest) to 10–40 mL/min, according to the limit of the microfluidic device. All other operating parameters were kept constant, with an S:AS volumetric ratio of 1:4, 100 mg/g ITZ in NMP as the feed solution concentration, 0.5% *w*/*w* Vit E TPGS 1000 aqueous solution as the antisolvent, a 40 mL volume of final suspension collected, an experimental and storage temperature of 25 °C, and an aging time of 48 h in the agitated crystallizer.

#### 2.4.3. Stabilizer Selection

The impact of the five most promising parenterally approved excipients for the stabilization of ITZ suspensions based on an earlier optimization study conducted on the MCR-CLASC process [54] was investigated in the SCT-CLASC process and compared against pure water without any stabilizer as a reference. A quantity of 1 g of surfactant, Vit E TPGS 1000, POL 188, POL 338, or POL 407 was dissolved in 199 g of Millipore water at 25 °C to create 0.5% *w*/*w* aqueous solutions. Similarly, polymers such as NaCMC and PVPK30 were produced as 0.5% *w*/*w* aqueous solutions. The solubility of ITZ in each of the 0.5% *w*/*w* aqueous stabilizer solutions and Vit E TPGS 1000 also, at 0.075% *w*/*w*, were conducted. An excess amount of ITZ was added to each 10 g of the 0.5% *w*/*w* aqueous stabilizer solutions, and the slurries were equilibrated for 72 h under stirring at 800 rpm at 25 °C. The ITZ concentrations were determined by chromatography analysis, as reported in the earlier work published by our group [54]. All solubility experiments were conducted in triplicate, and the standard deviations were reported accordingly.

All other operating parameters were kept constant, including the SCT-CLASC setup, the co-flow static mixer, the S–AS stream flow rate (5–20 mL/min), the S:AS volumetric ratio (1:4), the ITZ in NMP feed solution concentration (100 mg/g), the final volume of suspension collected (40 mL), the process and aging temperature (25 °C), and the 48 h aging period in the agitated crystallizer prior to PSD analysis. Following that, the SCT-CLASC process was carried out using each of the different aqueous stabilizer solutions as antisolvents. To create thick wet slurries for PXRD analysis, the generated suspensions were filtered using a Durapore^®^ 0.22 µm filter. This allowed for the identification of the solid-state form of ITZ in the suspensions.

#### 2.4.4. API Feed Solution Concentration

The effect of the ITZ feed solution concentration in the NMP on the resulting ITZ PSD, solid-state form, and morphology of the suspensions generated using the SCT-CLASC process was evaluated using a co-flow mixer as a mixing device. The tested ITZ feed solution concentrations were 10, 50, 100, and 200 mg/g solution and were prepared by dissolving 1, 5, 10, and 20 g of ITZ in 99, 95, 90, and 80 g of NMP, respectively, under constant stirring. All solutions were filtered with a Durapore^®^ 0.22 µm PVDF membrane (Merck Millipore Ltd., Cork, Ireland) prior to the continuous LASC process. All other operating parameters were kept constant, with an S:AS volumetric ratio of 1:4, an S–AS stream flow rate of 5–20 mL/min, and 0.5% *w*/*w* Vit E TPGS 1000 aqueous solution as antisolvent, with a 40 mL volume of final suspension collected, a storage temperature of 25 °C, and an aging time of 48 h in the agitated crystallizer. Afterward, a higher ITZ feed solution concentration of up to 400 mg/g was tried at 65 °C (the highest operational temperature in Secoya SCT-LAB instrument of Secoya Technologies (Louvain-la-Neuve, Belgium)) to keep the ITZ dissolved in the feed solution; however, the mixing device blocked above 200 mg/g due to fouling, encrustation, and back-mixing. As a result, no additional suspension was generated for the same.

#### 2.4.5. Microfluidic Mixing Device Comparison

The impact of various static mixing devices (e.g., a co-flow mixer, a T-cross mixer, and a Y mixer) against no mixing device, i.e., basically no connector (referred to as “no mixer”), was investigated in the Secoya SCT-LAB microfluidic setup. A 40 mL volume, temperature-controlled, agitated crystallizer was used as a collecting and aging vessel for the produced suspensions, which were collected via FEP outlet tubing. All other operating parameters were kept constant, with an S:AS volumetric ratio of 1:4, an S–AS stream flow rate of 5–20 mL/min, 100 mg/g ITZ in NMP as the feed solution concentration, 0.5% *w*/*w* Vit E TPGS 1000 aqueous solution as the antisolvent, a 40 mL volume of final suspension collected, an experimental and storage temperature of 25 °C, and an aging time of 48 h in the agitated crystallizer.

### 2.5. Optimization of Vit E TPGS 1000 Concentration

Vit E TPGS 1000 was pre-melted, and 20 g was dissolved in 380 g of water to make an aqueous solution of 5% *w*/*w*. The stock solution was then diluted 10, 16.7, 20, 25, 33.3, 50, 66.7, and 100 times with water to produce 100 g of Vit E TPGS aqueous solution with 0.5, 0.3, 0.25, 0.2, 0.15, 0.1, 0.075, and 0.05% *w*/*w* concentrations, respectively. Thereafter, the SCT-CLASC process, using a co-flow mixer as a static mixing device for each of the antisolvent solutions, was carried out both at medium S–AS stream flow rates (5–20 mL/min) and high S–AS stream flow rates (10–40 mL/min), as described in Section 2.4.2, keeping all other operating parameters constant. The effect of various Vit E TPGS 1000 concentrations in different mixing devices in the microfluidic setup on the resulting PSD, solid-state form, and morphology of the ITZ crystals in the post-precipitation suspensions (for solid loadings of 20 mg/g and 40 mg/g) after 48 h of aging in the agitated crystallizer at 25 °C was evaluated by laser diffraction for PSD in quadruplicate, along with PXRD and SEM analysis.

### 2.6. Downstream Processing of LAI Suspensions

For the generation of a highly concentrated LAI suspension, the aged ITZ suspension in the crystallizer which was produced via the SCT-CLASC process was then subjected to an inline membrane diafiltration process, which was previously developed by Anjum et al. [22]. The ITZ solid loading in the final suspension was increased (via concentrating from either 20 mg/g or 40 mg/g ITZ loading in the post-precipitation suspension to the final theoretical solid loading of 300 mg/g) together with the residual solvent (NMP) removal by diafiltration using a dead-end membrane cell (with parts from Gillain & Co, Aartselaar, Belgium and Swagelok, OH, USA). The integrated separation and concentration process was directly adapted from the literature [55]. After that, a homogenous LAI suspension with a solid loading of 300 ± 10 mg/g was achieved at 25 °C by magnetic stirring at 250 rpm after the ITZ wet pellets above the membrane were reconstituted in the necessary amount of 0.5% *w*/*w* Vit E TPGS 1000 aqueous solution as a resuspending agent. For the feed suspension made with 0.075% *w*/*w* Vit E TPGS 1000 aqueous solution as an antisolvent, the same process was followed, and the ITZ wet pellets were resuspended using 0.075% *w*/*w* Vit E TPGS 1000 aqueous solution. SEM imaging was performed, and laser diffraction for PSD and PXRD (to verify the polymorphic forms in the suspensions) was used to analyze all three of the LAI suspensions at a 300 mg/g ITZ concentration.

### 2.7. Stability Study

To investigate the stability of the four optimized post-precipitation suspensions (F1.1: suspension with a solid loading of 20 mg/g, Vit E TPGS 1000 0.5% *w*/*w*; F1.2: suspension with a solid loading of 20 mg/g, Vit E TPGS 1000 0.075% *w*/*w*; F2.1: suspension with a solid loading of 40 mg/g, Vit E TPGS 1000 0.5% *w*/*w*; F2.2: suspension with a solid loading of 40 mg/g, Vit E TPGS 1000 0.075% *w*/*w*) generated by the SCT-CLASC process and four subsequent LAI microsuspensions over time, samples were stored in unstirred conditions in sealed glass vials in controlled stability chambers at 25 °C for 4 months. The physical stability of the suspensions was measured in terms of resuspend ability. After this period, there should not be any sediment at the bottom of the vial after 30 s of swirling. In addition, PSD and crystallinity (determined by PXRD) were assessed for all optimized suspensions (both feed and LAI suspensions) after ensuring the homogeneity of the suspensions.

### 2.8. Particle Size Distribution (PSD)

A laser diffraction particle size analyzer (Mastersizer MS3000, Malvern Instruments Ltd., Malvern, UK) fitted with an automatic dispersion unit, the Hydro 3000S, utilizing water as the dispersion medium, was used to calculate the PSDs (D10, D50, and D90) of the ITZ suspensions. The dispersant refractive index was 1.333, and the refractive index and absorption values for ITZ were 1.612 and 0.01, respectively. A degree of obscuration between 14% and 20% was achieved by dispersing each suspension in purified water while maintaining a stirring velocity of 2500 rpm for the water medium. Every measurement was conducted with a 30 s pre-measurement delay. A 100 s pre-measurement sonication was used for the suspensions prepared from 200 mg/g ITZ in NMP. Using Mie theory, particle sizes were determined and are shown as volume-based median equivalent sphere diameters. Every measurement was carried out in triplicate.

### 2.9. Solid-State Characterization

#### 2.9.1. Scanning Electron Microscopy (SEM)

The morphological features of ITZ crystals were examined using PHENOM^™^PRO (PHENOMWORLD BV, Noord-Brabant, CA, USA). Samples were prepared by spreading a small droplet of ITZ suspension onto an adhesive carbon tape previously attached to a 15 mm cylindrical aluminum SEM stub. After air drying, the samples were coated with gold using an Emitech K550 (Emitech, Ashford, UK) sputter coater for 60 s at 20 mA. The coated samples were then examined at an acceleration voltage of 10 kV, a 30 μm aperture, and an average working distance of 10 mm. At least 10 images were captured for each sample to verify its consistency. All SEM images presented in this research were fully representative of each sample analyzed.

#### 2.9.2. Powder X-ray Diffraction (PXRD)

PXRD was used to identify the polymorphic form and monitor the degree of crystallinity of the feed suspensions generated by the SCT-CLASC process and the LAI suspensions after downstream processing. To obtain wet mass values and ensure the determination of the polymorphic forms present in the suspensions, the produced suspensions were filtered through a Buchner filtering setup using a Durapore^®^ 0.22 µm PVDF membrane (Merck Millipore Ltd., Cork, Ireland). Subsequently, the wet paste samples were placed on “zero background” holders just before measurement and run on an Aeris benchtop powder diffractometer (Malvern Panalytical, Malvern, UK) using non-monochromated Cu Kα radiation. The voltage used was 40 kV, and the current was 15 mA. The measurements covered the range of 4–50° 2θ, with a 0.0217° step size. The diffraction data were processed using the X’Pert HighScore PlusR software v2.2a (PANalytical BV).

### 2.10. Residual Solvent Analysis

Gas chromatography (GC) with flame ionization detection (FID) was used to determine the concentrations of NMP in the feed suspensions, diafiltration retentates, and concentrated suspensions. An Agilent 7890/6890 device (Agilent Technologies, Santa Clara, CA, USA) with an Rxi 624 Sil MS column (Restek, Bad Homburg vor der Höhe, Germany) was used. Helium, with a flow rate of 1.0 mL/min, was used as the carrier gas. A combination of hydrogen and synthetic air was used as the detector gas. After programming the oven temperature gradient, the samples were run for 10 min. The GC analyses were preceded by a sample preparation step in which the samples were filtered through 0.45 µm PTFE Acrodisc^TM^ syringe filters (Pall Corporation, Fajardo, Puerto Rico).

### 2.11. Vit E TPGS 1000 Content Analysis

The concentration of Vit E TPGS 1000 in the post-precipitation suspensions and concentrated LAI suspensions was determined using an Acquity ultra-high-performance liquid chromatography (UPLC) system with a photodiode array (PDA) detector (Waters, Milford, CT, USA). The stationary phase consisted of an Ethylene Bridged Hybrid (BEH) C18 column with a particle size of 1.7 µm. The mobile phase consisted of 40% *v*/*v* acetonitrile and 60% *v*/*v* purified water. Isocratic elution was applied during the experiments. The flow rate was set at 0.5 mL/min, and the temperature was set at 45 °C. Per sample, 1.0 µL was injected into the column, which ran for 6 min. UV measurements were performed at a wavelength of 220 nm. 

Samples were prepared by dissolving 50 µL of suspension in 950 µL of NMP for feed suspensions and 10 µL of suspension in 990 µL of NMP for concentrated LAI suspensions. The content of Vit E TPGS 1000 present in the suspensions was determined from the UPLC analysis.

### 2.12. Itraconazole Loading in Final LAI Suspensions

To determine the concentration of ITZ in the suspensions, a gravimetric analysis was performed. After weighing, the samples were placed in a Heraeus Vacutherm VT 6060M drying chamber (Heraeus, Hanau, Germany) at 80 °C under a vacuum for 96 h. Afterward, the samples were reweighed to determine the weight percentage of ITZ. This was corrected to account for the Vit E TPGS 1000 present. This method was performed on the feed suspensions after the SCT-CLASC process and the LAI suspensions after the centrifugal filtration process.

### 2.13. In Vitro Discriminatory Dissolution Comparison

The dissolution profiles of three optimized stable ITZ LAI suspensions, i.e., the ITZ LAI suspension denoted F3|MCR-CLASC (PSD: D10 1.2, D50 3.4, and D90 6.9 µm), generated via downstream processing from a post-precipitation suspension loading of 10 mg/g after the MCR-CLASC process, with Vit E TPGS 0.5% *w*/*w* as a stabilizer; the ITZ LAI suspension denoted F4|SCT-CLASC (PSD: D10 1.5, D50 3.1, and D90 6.1 µm), generated via downstream processing from post-precipitation suspension loading of 40 mg/g after the SCT-CLASC process, with Vit E TPGS 1000 0.5 % *w*/*w* as a stabilizer; and the ITZ LAI suspension denoted F6|SCT-CLASC (PSD: D10 1.5, D50 4.1, and D90 9.8 µm), generated via downstream processing from a post-precipitation suspension loading of 40 mg/g after the SCT-CLASC process, with Vit E TPGS 0.075% *w*/*w* as a stabilizer, were evaluated and compared against the dissolution profile of the ITZ suspension, which comprised the as-received API suspension (PSD: D10 5.5, D50 15.2, and D90 42 µm) in Vit E TPGS 1000 0.5% *w*/*w* medium. 

The dissolution profiles were studied in 500 mL of 1% *w*/*v* Tween 20 + 0.1N HCl (pH 1.2) release medium at 37 ± 0.5 °C in a temperature-controlled water bath with submerged stirring at 300 rpm, as described in the earlier work published by our group [54]. This medium was selected as the discriminatory medium for testing differences in the properties of the ITZ LAI suspensions, considering the impact of particle size and ensuring complete release. Sink conditions were used for these experiments, with a maximum concentration of API of 5 mg/L. The concentration of dissolved drug was measured using a Shimadzu UV-1800 UV-Vis spectrophotometer, with λ_max_ = 261 nm (ITZ), by withdrawing an aliquot from the solution with a pre-heated plastic syringe after pre-set time intervals (i.e., 1, 3, 5, 10, 15, 20, 30, 60, 120, 240, 360, 480, 1440, 2880, and 4320 min) and filtering using pre-heated PTFE syringe filters with a pore size of 0.22 μm. The measurements were carried out in triplicate.

## 3. Results and Discussion

### 3.1. MCR-CLASC Process

The MCR-CLASC setup described in this study, as well as the operating parameters used to generate a stable ITZ suspension using a Y mixer in a microchannel reactor-based CLASC process, were optimized previously by our group [54]. Therefore, 100 mg/g ITZ in NMP and 0.5% *w*/*w* Vit E TPGS 1000 aqueous solution were used as the S:AS combination; the experiments were performed at 25 °C and kept at the same temperature during storage. For further comparison with the suspensions generated by the SCT-CLASC process, the ITZ PSDs for the S:AS ratios of 1:4, 1:5, and 1:8 are illustrated in Figure 2A. 

Figure 2A shows that only with an S:AS ratio of 1:8 was a PSD range with a D90 value below 10 µm achieved; therefore, this was chosen as the optimal S:AS ratio using the Y mixer. Lower S:AS ratios, such as 1:4, resulted in a wider PSD and a shift towards a higher D50 value, with a D90 above 10 µm.

The particle morphology of the ITZ particles in the suspensions was analyzed by SEM, and the results are shown in Figure 2B, C for S:AS ratios 1:4 and 1:8, respectively. It was observed that the crystalline ITZ particles presented an elongated tabular structure with a smooth surface and a nearly uniform shape. No difference in morphology between the two ratios was observed, apart from a difference in the particle size. 

### 3.2. Secoya Microfluidic Crystallization Technology-Based Continuous Liquid Antisolvent Crystallization (SCT-CLASC) Process

#### 3.2.1. Operating Parameter Optimization

##### S:AS Ratio

To study the effect of the S:AS ratio on ITZ PSDs after the SCT-CLASC process using co-flow and T-cross static mixers, this parameter was varied from 1:3 to 1:8, as illustrated in Figure 3. The other operating parameters were kept constant, as mentioned in Section 2.4.1. The first operating parameter to be optimized was the S:AS ratio using the co-flow mixing device. Since S:AS ratios of 1:1 and 1:2 caused the mixing device to become clogged due to the limited fluid velocity, as the driving forces for crystallization were lower in these cases, they were discarded from further experiments. 

It can be observed from Figure 3 for the co-flow mixer that S:AS ratios of 1:4 or higher (up to 1:8) originated suspensions with PSDs within the target (1–10 µm) limit. The D50 values changed from 3.8 ± 0.4 to 2.9 ± 0.3 µm, while the D90 values showed a similar trend, decreasing slightly from 6.9 ± 0.8 to 5.1 ± 0.3 µm. For the S:AS ratio of 1:3, the D50 value was 7.3 ± 0.6 µm and the D90 value was 18.8 ± 1.2 µm, which fell outside the intended range. For the T-cross mixer, it was observed that only an S:AS ratio of 1:5 resulted in a PSD range below 10 µm, where the D50 and D90 values were 3.8 ± 0.5 and 8.3 ± 0.7 µm, respectively. For an S:AS ratio of 1:4, the D50 and D90 values were 6.3 ± 0.3 and 13.8 ± 0.7 µm, while for an S:AS ratio of 1:8, the D50 and D90 values were 3.8 ± 0.8 and 16.5 ± 0.4 µm, respectively, with a multimodal distribution. T-cross suspensions tend to present a broader particle size distribution, as a follow-up from the explanations of both mixers.

In a co-flow mixer, the solvent from the inner microchannel tends to expand into the outer antisolvent layer due to diffusive mixing, and nuclei formation occurs at the interface between the solvent and the antisolvent. This means that it migrates directly to its equilibrium position with rapid depletion of the supersaturation gradient. Consequently, a reduction in the solvent-to-antisolvent ratio (S:AS) leads to a reduction in the equilibrium solubility of the drug, followed by high supersaturation, which results in a particle size decrease. Furthermore, due to high-intensity mixing inside the co-flow and tubular crystallizer and control over crystal growth, due to the presence of the stabilizer, the PSD almost reached a plateau phase. Since the S:AS ratio of 1:4 had the highest final solid loading of API in suspension, it was selected as the optimal S:AS ratio for further studies. 

In a T-cross mixer, both liquid streams collide perpendicularly, causing turbulent mixing of the solvent and the antisolvent. This usually results in the formation of many small nuclei with a large surface area. In the present study, due to the higher agglomeration of particles, this resulted in a lack of a clear trend for the system, such that large variability in the sizes of the particles generated in the T-cross mixer was observed. It could also be possible that when using the S:AS ratio of 1:8 in the T-cross mixer, the flow rate difference was too large, which caused back-mixing at the outlet, resulting in a broader PSD [52]. Therefore, an S:AS ratio of 1:5 was selected as optimal when using the T-cross mixer for further process parameter optimization studies.

Appendix A show that the obtained particles had an elongated tabular structure with irregular edges but smooth surfaces. Particle morphology appeared to be similar for both conditions, with S:AS ratios of 1:4 and 1:8 using the co-flow mixer. In Appendix A, SEM images of ITZ crystals show a rod-shaped morphology with irregular edges and a smooth surface for both S:AS ratios in the T-cross mixer. Larger particles can be observed in the suspensions generated using the T-cross mixer.

According to Appendix A, the PSDs for all solvent and antisolvent volumetric (S–AS) stream flow rates were found to be similar and all were within the target PSD interval of 1–10 µm. However, with the S–AS stream flow rate of 1–4 mL/min, fouling and encrustation were observed in the tubular crystallizer portion, which disrupted the continuous operation. The highest S–AS stream flow rates of 10–40 mL/min were too close to the operational limits (45–45 mL/min) of the SCT-CLASC system. Regardless of the S–AS flow rate, the morphology of the ITZ particles remained unchanged, as shown in Appendix A. As a result, the S–AS flow rate of 5–20 mL/min at 25 °C with an S:AS ratio of 1:4 was determined to be optimal in terms of the PSD, solid-state form, and morphology of the ITZ crystals in the suspensions.

##### Stabilizer Selection

To retain particle size, inhibit crystal growth, prevent agglomeration of precipitated particles, and stabilize a suspension, an excipient, such as a polymer (which provides steric stabilization), a surfactant (which provides electrostatic stabilization), or a combination of both, is added to the suspension [56]. Additionally, stabilizers can lower the free energy barrier for nucleation by lowering the solution’s surface tension [36,56,57]. 

The effect of stabilizer selection on the resulting PSDs and the solid states of the produced suspensions was investigated, while other operating parameters were kept constant, as described in Section 2.4.3. Based on earlier work carried out by our group [54], six stabilizers commonly used for parental applications were selected and compared against an ITZ suspension produced in water without any excipient as a reference, as demonstrated in Figure 4. 

The results for the equilibrium solubility of ITZ in various stabilizer solutions are shown in Table 1. It can be observed that the solubility increases significantly from ng/g to µg/g level when compared with pure water [54], probably due to the solubilization effect of the stabilizers. The equilibrium solubility for the six investigated surfactants varies between 2 and 13 µg/g solution at 25 °C, which is still in a very poorly soluble range, while in the case of the antisolvent crystallization process, the ITZ feed solution concentration is several folds higher at 100 mg/g solution. The highest solubility for ITZ was obtained in the Vit E TPGS 1000 solution, followed by POL 407 at a similar concentration. ITZ solubility in an aqueous solution of Vit E TPGS 1000 decreases with a lower stabilizer concentration in the medium.

The suspension generated in water without any excipient had the largest PSD of the ITZ particles, with D50 and D90 values of 17.5 ± 1.5 and 59 ± 2.5 µm, respectively. The largest particle size in water, as a reference, confirmed the need for a stabilizer in the suspension to reduce particle growth and agglomeration. The D50 values for the ITZ particle size for the six investigated surfactants decreased as follows: PVP K30 (10.5 ± 1.1 µm) > POL 338 (10.5 ± 0.6 µm) > POL 407 (9.2 ± 0.7 µm) > NaCMC (9.2 ± 0.5 µm) > POL 188 (8.1 ± 0.3 µm) > Vit E TPGS 1000 (3.3 ± 0.3 µm), while the D90 values for the ITZ particles decreased as follows: NaCMC (84.2 ± 8.2 µm) > POL 338 (35.4 ± 1.6 µm) > POL 407 (31.2 ± 1.2 µm) > PVP K30 (28.2 ± 1.5 µm) > POL 188 (25.1 ± 0.8 µm) > Vit E TPGS 1000 (6.3 ± 0.5 µm), as illustrated in Figure 4. Among these, a stable ITZ suspension with the narrowest PSD within the target range (1–10 µm) was achieved with Vit E TPGS 1000, which is consistent with the results reported for the MCR-CLASC procedure [54]. The presence of a stabilizer, which preferentially adsorbs on the particle surface, further inhibited particle development via steric and/or electrostatic stabilization [38]. When compared to other hydrophilic polymers or surfactants, amphiphilic surfactants such as Vit E TPGS 1000 have been reported in the literature to be more efficient in stabilizing ITZ microparticles by forming a protective layer around the particles, decreasing the interfacial tension and increasing the wettability of ITZ crystals in suspension [58]. In conclusion, Vit E TPGS 1000 was shown to be the most promising excipient for stabilizing the produced suspensions.

The main characteristic peaks of ITZ in the PXRD patterns in Appendix A were consistent with the standard data of the as-received ITZ API, indicating that the stable crystal form I did not change before or after the SCT-CLASC process when stabilizers were used.

The micrograph in Figure 5A shows an elongated rod-shaped structure with a non-uniform particle size in the case of the as-received ITZ API. The ITZ particles in water, as a reference, shown in Figure 5B, without any excipient, showed a tabular morphology with a smooth surface but irregular edges.

The morphologies of the ITZ particles generated with six different stabilizers were Figure 5C rod-shaped with rough edges, Figure 5D elongated-rod-shaped with a rough surface, Figure 5E thin-rod-shaped but agglomerated, Figure 5F thin rods stacked upon each other in a hay-like formation, Figure 5G needle-like in shape, and Figure 5H uniformly sized elongated tabular structures for POL 188, POL 338, POL 407, PVP K30, NaCMC, and Vit E TPGS 1000, respectively. The variety of morphologies obtained showed that the type of excipient had an influence on the habit formation of the ITZ in suspension [59,60]. Therefore, it is critical to understand how excipients influence ITZ crystallization, which can provide experimental support for rational excipient selection for suspension optimization [61] during the SCT-CLASC process. Hence, based upon evaluating the PSDs, solid-state forms, and particle morphologies in the suspensions after aging, Vit E TPGS 1000 was selected as the optimal stabilizer for ITZ suspensions.

The PSDs obtained were not significantly different between the different tested ITZ feed solution concentrations, as shown in Appendix A. The suspension generated from 200 mg/g ITZ feed solution in NMP still achieved the target PSD, with a D50 value of 3.8 ± 0.4 µm and a D90 value of 8.2 ± 0.9 µm. Since this was the highest concentration that still met the PSD criteria, it was selected as the optimal ITZ feed solution concentration for future experiments in LAI suspension generation.

##### Effects of Static Mixing Devices on the SCT-CLASC Process

Three static mixing devices, co-flow, T-cross, and Y mixers, were examined against no mixing device, i.e., basically no connector (referred to as “no mixer”), in the same microfluidic system to study the effect of mixing intensity on the production of ITZ crystal suspensions using the SCT-CLASC procedure. Other operating parameters were kept constant, as described in Section 2.4.5. To understand the influence of the static mixers on ITZ nuclei formation, the PSDs and morphologies of the ITZ suspensions produced by the co-flow, T-cross, and Y mixers were compared to those produced with no mixer just after precipitation, after 10 min, and after 48 h by the SCT-CLASC method.

Figure 6 (time: 10 min) shows that nanoparticles were formed in each case for ITZ, independently of whether no mixing device or a static mixing device was used. The D50 values for the initial ITZ nuclei were 0.1 ± 0.2, 0.2 ± 0.8, 0.3 ± 1.0, and 0.6 ± 1.2 µm, while the D90 values were 0.5 ± 0.2, 0.8 ± 0.3, 1.6 ± 0.6, and 3.5 ± 0.9 µm, when the mixer condition was changed from no mixer to the co-flow, T-cross, and Y mixer conditions, respectively. The PSD obtained with no mixer appears to have the narrowest distribution, while the PSD obtained with the T-cross mixer appears to have the widest distribution, probably due to rapid agglomeration just after mixing. After 48 h, it can be observed that all four suspensions were in the micron size range. The D50 values of the aged ITZ particles increased, these being 3.8 ± 1.2, 3.3 ± 0.5, 6.3 ± 0.5, and 6.3 ± 0.7 µm, while the D90 values were 493 ± 26.7, 6.6 ± 0.4, 12.5 ± 0.5, and 17.6 ± 0.3 µm, when the mixer condition was changed from no mixer to the co-flow, T-cross, and Y mixer conditions, respectively. Figure 6 (time: 48 h) shows that when the T-cross and Y mixers were employed, the PSDs were broader and shifted to a higher value than when the co-flow mixer was used. Furthermore, the PSD obtained with no mixer displayed the widest multimodal distribution, with an additional peak in the 100–1000 µm size range due to agglomeration and uncontrolled crystal growth [39]. This observation contributes towards confirmation of the hypothesis presented before, namely, that the PSD was primarily controlled by Vit E TPGS 1000.

In addition, the SEM images in Figure 6 (time: 10 min) show spherical globular particles in the (sub-)micron range, which is indicative of the meso-crystalline form of ITZ. The appearance of the tabular or rod-shaped structures in the last three micrographs indicates a rapid transformation from the metastable meso-crystalline form to the stable form I of ITZ, due to the high-intensity mixing in the static mixer compared with no device. Mugheirbi and Kozyra et al. [62,63] have previously described the emergence of the liquid crystalline or meso-crystalline form after the LASC process and three distinct crystal habits of ITZ after the LASC process, in addition to three distinct crystal habits of ITZ and an amorphous form, which is in accordance with what was observed in this work. After 48 h, the morphologies of the crystalline suspensions of ITZ prepared using no mixer, the co-flow mixer, the T-cross mixer, and the Y mixer were needle-like aggregated structures of non-uniform size; elongated tabular structures; irregular, rod-shaped structures; and irregular porous plates with rough surfaces, respectively. These observations confirmed that mixing has an impact on ITZ habit formation. Since the co-flow mixer had the smallest median value and the narrowest PSD, it was selected as the optimal mixing device for the SCT-CLASC process.

The main characteristic peaks in the PXRD patterns of the ITZ suspensions produced with the different mixing devices shown in Appendix A were consistent with the standard data of the as-received ITZ API, indicating that the stable crystal form did not change before or after the SCT-CLASC process independently of the chosen mixing device.

### 3.3. Stabilizer Concentration Optimization

Following the optimization of the majority of the SCT-CLASC process operating variables, the optimal stabilizer concentration needed to stabilize ITZ crystals in suspension was determined. This was achieved by comparing the performance of various % *w*/*w* Vit E TPGS 1000 aqueous feed solutions as antisolvents to that of reference water without any stabilizer while using the co-flow mixer as a static mixer at a medium optimized S–AS flow rate of 5–20 mL/min. Other process variables remained constant, including the S:AS ratio of 1:4 and the 100 mg/g ITZ feed solution concentration, under storage in an agitated crystallizer for 48 h at 25 °C. The effect of reducing the Vit E TPGS 1000 concentration in the antisolvent on the PSDs of the produced suspensions is shown in Figure 7. 

Figure 7A,B show the impact of varying stabilizer concentrations on the PSDs of ITZ suspensions. In Figure 7A, it can be observed that by reducing the Vit E TPGS 1000 concentration from 0.5, 0.3, 0.2, 0.1, and 0.075 to 0.05% *w*/*w*, the D50 values varied slightly between 3.3 ± 0.5 and 4.9 ± 0.3 µm without showing any clear trend, then substantially increased to 17.5 ± 2.7 µm for 0% *w*/*w*. Similarly, the D90 values varied between 6.9 ± 0.5 and 10.2 ± 0.4 µm and increased to 59 ± 2.3 µm for 0% *w*/*w*. It is clear that, without a stabilizer, the suspensions’ stability was jeopardized, as observed in Section Effect of Static Mixing Device on SCT-CLASC Process. The uncontrolled crystal growth caused by Ostwald ripening resulted in significantly higher PSDs than that obtained with the initial raw ITZ API that was used. Moreover, all tested concentrations produced desired PSDs, the values falling within the target range, with minor variations based on experimental observation of both the D50 and D90 values. Between the concentrations of 0.075 and 0.05% *w*/*w*, a narrower distribution with lower D50 and D90 values was obtained for the Vit E TPGS 1000 concentration of 0.075% *w*/*w* during repeated experiments. Given the later stages of process intensification and concentration, 0.075% *w*/*w* was selected as the optimal concentration of Vit E TPGS 1000 for stabilizing ITZ microsuspensions in the SCT-CLASC process. Following the optimization of the majority of the SCT-CLASC process operating variables, the impact of varying stabilizer concentrations on the quality of the generated ITZ suspensions was evaluated using the co-flow mixer at a high S–AS flow rate of 10–40 mL/min (Figure 7B). Other process variables remained constant, as mentioned in Section 2.5.

As can be observed in Figure 7B, when the Vit E TPGS 1000 concentration was reduced from 0.5 and 0.1 to 0.075, the D50 values increased only marginally from 3.8 ± 0.3 to 4.3 ± 0.2 µm, then substantially to 9.2 ± 0.5 µm for 0.05% *w*/*w*, while the D90 values increased from 7.4 ± 0.5 and 9.8 ± 0.6 to 12.4 ± 0.8 µm and then again to 72 ± 2.4 µm, with a bimodal distribution for 0.05% *w*/*w*. 

According to Figure 7A,B, a decrease in stabilizer content in the suspension was correlated with a tendency toward a wider size distribution and size shifting to a higher range. For concentrations of 0.075% *w*/*w* or higher of Vit E TPGS 1000, a stable PSD within the limits of the target range for LAI suspensions was obtained. 

In the co-flow mixer condition, the time for which the solvent and the antisolvent remained in contact depended on the difference in their flow velocities. Therefore, at very low flow rates, particle migration toward its equilibrium position was slower, allowing sufficient time for crystal growth inside the mixer. Similarly, at higher flow rates (Figure 7B), particle migration was fast enough to result in a non-uniform PSD. The balance was observed at the medium flow rate (Figure 7A), and experimental observations showed that the flow rate played a less critical role in determining the final PSDs of the suspensions produced by the co-flow mixer. The PSDs obtained were wider and shifted to higher values for S–AS flow rates of 10–40 mL/min compared to S–AS flow rates of 5–20 mL/min. It is possible that at a higher flow rate in a co-flow mixer, due to very rapid diffusive mixing and a shorter residence time, non-uniform depletion of the supersaturation gradient may induce non-homogeneous crystal formation with a very small variance. Furthermore, the concentration of Vit E TPGS 1000 at or below 0.05% *w*/*w* was insufficient to successfully stabilize the ITZ particles in the suspensions. As a result, 0.075% *w*/*w* Vit E TPGS 1000 aqueous solution again proved to be optimal as an antisolvent even at higher flow rates. The morphologies of the ITZ crystals in the suspensions derived from four different Vit E TPGS 1000 concentrations obtained from SEM measurements are presented in Appendix A. No significant difference in morphology was observed. It could be concluded that the ITZ crystals had elongated tubular structures with irregular edges and a smooth surface for all four Vit E TPGS 1000 concentrations. 

### 3.4. Comparison of Optimized Suspensions after the SCT-CLASC Process

After optimization of all operating process parameters of the SCT-CLASC process for ITZ, four selected suspensions were generated and compared in terms of their PSDs, solid states, and morphologies to assess the suspensions’ quality. These were produced using the co-flow mixer as a static mixing device with an S:AS ratio of 1:4, an S–AS flow rate of 5–20 mL/min at 25 °C, and aging for 48 h in an agitated crystallizer at 25 °C. Two of the suspensions were made with 100 mg/g ITZ feed solution and 0.5% or 0.075% *w*/*w* Vit E TPGS 1000 aqueous solution as the antisolvent, corresponding to 20 mg/g of theoretical solid loading in the suspensions after the SCT-CLASC process. The other two were produced using 200 mg/g ITZ feed solution and 0.5% or 0.075% *w*/*w* Vit E TPGS 1000 aqueous solution as the antisolvent, corresponding to 40 mg/g of theoretical solid loading in the suspensions after the SCT-CLASC process. Thus, the solid loading of ITZ was maximized while keeping the Vit E TPGS 1000 concentration as low as possible to ensure that the target PSD range (1–10 µm) was obtained with a D90 value below 10 µm. 

As observed in Figure 8, the PSDs of ITZ in all of the produced suspensions were within the target range. The D50 values remained almost similar, with only slight variations between 3.3 ± 0.5 and 4.3 ± 0.3 µm; similarly, the D90 values varied between 6.3 ± 0.5 and 9.9 ± 0.4 µm for the four tested ITZ suspensions. The PSD of the ITZ suspension increased slightly with increasing solid loading, while the Vit E TPGS 1000 concentration in the final suspension decreased, probably due to crystal growth. These results were consistent with the trend observed during the process parameter optimization.

PXRD was used to examine the solid-state forms of the four optimized suspensions, and the patterns are shown in Appendix A. The ITZ crystals in all four optimized suspensions had a solid-state form I, which remained unchanged when compared against the reference raw material (ITZ as received). SEM was used to characterize the morphologies of the ITZ crystals of the four optimized suspensions. The results are presented in Appendix A. All SEM images of the four optimized suspensions show an elongated plate-shaped morphology with a smooth surface and irregular edges of the ITZ. No significant difference was observed between the micrographs.

### 3.5. Downstream Processing of LAI Suspensions

#### 3.5.1. PSD Comparison of LAI Suspensions

Downstream processing via centrifugal filtration was required to concentrate the feed suspensions from the SCT-CLASC process, followed by resuspension of the centrifugate in the same stabilizer aqueous medium used earlier as the antisolvent for the respective feed suspensions. As a result, a larger quantity of concentrated LAI suspension can be prepared in less time, with fewer centrifugal devices and less energy being used due to the concentration being carried out in a single phase. The three chosen post-precipitation feed suspensions obtained via the optimized SCT-CLASC process were: F4|SCT-CLASC (i.e., post-precipitation suspension with a solid loading of 40 mg/g and Vit E TPGS 1000 0.5% *w*/*w* as medium); F5|SCT-CLASC (i.e., post-precipitation suspension with a solid loading of 20 mg/g and Vit E TPGS 1000 0.075% *w*/*w* as medium); and F6|SCT-CLASC (i.e., post-precipitation suspension with a solid loading of 40 mg/g and Vit E TPGS 1000 0.075% *w*/*w* as medium). These three suspensions were selected to evaluate the effect of higher solid loading (from 20 to 40 mg/g) and decreasing stabilizer concentration (from 0.5 to 0.075% *w*/*w*, corresponding to API-to-stabilizer ratios of 8:1 and 53:1) on product quality following downstream processing. They were then compared to the LAI suspension obtained via the previously optimized MCR-CLASC process, designated F3|MCR-CLASC (post-precipitation suspension with a solid loading of 10 mg/g and Vit E TPGS 1000 0.5% *w*/*w* as medium, with an API-to-stabilizer ratio of 2:1). In Figure 9, the PSDs of four ITZ LAI microsuspensions (obtained from feed suspensions with API-to-stabilizer ratios of 2:1, 8:1, 26:1, and 53:1, respectively) with a theoretical final ITZ solid loading of 300 mg/g are presented. 

All PSDs were within the target range of 1–10 µm. There were no direct indications of aggregation or crystal growth. In addition, no ITZ crystals were observed visually with the naked eye for all tested LAI suspensions. When compared to the MCR-CLASC process, the SCT-CLASC process produced a suspension with four times higher solid loading and, as a result, reduced the amount of Vit E TPGS 1000 utilized by up to 26 times while retaining the desired product quality. This finding indicates that simplifying process intensification (requiring concentration by up to 8 times rather than 30 times or more to achieve an LAI concentration range) leads to waste reduction, cost reduction, and minimization of undesirable effects of excess stabilizers in terms of processing and physiology.

The PXRD analysis in Appendix A confirmed the presence of form I of ITZ in all four generated LAI suspensions, which remained unchanged when compared with the reference raw material. 

The SEM micrographs in Appendix A show a uniformly sized, elongated, plated structure with a smooth surface, although variation in crystal width was observed. These findings provided additional evidence that habitual ITZ crystals remained stable in LAI suspensions following downstream processing, which is in accordance with the findings of earlier work by Anjum et al. [55].

#### 3.5.2. Analysis of Final ITZ and Stabilizer Loading and Residual Solvent

After downstream processing (membrane diafiltration and reconstitution), final ITZ solid loading, Vit E TPGS 1000, and residual NMP concentrations in the final four LAI suspensions were determined using UPLC in triplicate, as described in Section 2.10, Section 2.11 and Section 2.12.

A detailed summary of the results is provided in Table 2. Average solid loadings of 29 ± 1.8, 30 ± 2.1, 29 ± 2.4, and 30 ± 1.6 ITZ were achieved in the LAI suspensions, i.e., F3|MCR CLASC, F4|SCT-CLASC, F5|SCT-CLASC, and F6|SCT-CLASC, respectively. This was in line with the target theoretical loading of 300 mg/g, demonstrating the potential of the centrifugal filtration method as an effective concentration strategy. By contrast, the concentrations of Vit E TPGS 1000 in the four abovementioned LAI suspensions were approximately 14.4 ± 0.6, 3.7 ± 0.1, 1.1 ± 0.1, and 0.5 ± 0.1, respectively. 

After solvent switch through the membrane-based diafiltration process described previously by Anjum et al. [55] and followed by redispersion in the fresh antisolvent medium, which was the vehicle for injection, residual NMP concentrations in the final reconstituted ITZ LAI suspensions of less than 0.05% *w*/*w* were obtained for all four tested LAI suspensions. Again, these values comply with ICH criteria [64]. These findings for ITZ indicate that the SCT-CLASC process, when paired with the membrane-based diafiltration (solvent separation and concentration) and reconstitution steps, constitutes an effective integrated bottom-up closed-loop process that can be effectively applied in an industrial environment to generate the intended outcomes for LAI suspensions.

### 3.6. Stability

The evaluation of the storage stability of microsuspensions is crucial for assessing final suspension quality and is of critical importance for LAI formulations with very high solid loadings of up to 300 mg/g, as finely dispersed particles in an aqueous stabilizer medium have a high tendency to agglomerate, leading to the formation of larger aggregates. Furthermore, Ostwald ripening (the process of larger particles growing at the expense of smaller particles due to the enhanced solubility of the latter) provided an additional stability challenge. As a result, the resuspendability, PSDs, and solid states of all three feed suspensions obtained via the optimized SCT-CLASC method and their corresponding LAI suspensions, as indicated in Section 3.5.2, were studied.

In terms of physical stability, all three feed suspensions obtained via the SCT-CLASC process, as well as the corresponding LAI suspensions, were resuspendable and homogeneous after 30 s of shaking when visually observed after 120 and up to 150 days of storage in a stability chamber at 25 °C. During stable storage at 25 °C, no hard cake formation was detected.

In Appendix A, the PSDs of all three feed suspensions generated by the optimized SCT-CLASC process (i.e., F1.2, F2.1, and F2.2), after 7 days and 150 days of storage are presented. The results show that, throughout the time of storage, the D90 values for all three suspensions were below 10 µm. However, at both measured time intervals, there was a slight decrease in particle size, while the PSD curve widened on either side. 

The stable form I of ITZ remained unchanged 150 days after the SCT-CLASC process when compared to the reference raw material, and PXRD studies revealed no change in crystallinity, as shown in Appendix A.

Figure 10 shows the PSDs of all three corresponding LAI suspensions of ITZ (i.e., F4, F5, and F6) with a final solid loading of 300 mg/g, produced after downstream processing of three post-precipitation suspensions, after 7 and 120 days of storage at 25 °C. 

The D10 values differed between 0.8 ± 0.1 and 1.6 ± 0.2 µm, the D50 values varied slightly between 2.9 ± 0.4 and 4.3 ± 0.3 µm, and the D90 values changed between 9.1 ± 0.7 and 11.2 ± 0.5 µm for the three LAI suspensions. The relative standard deviation (RSD) for the D10 values after 120 days of storage was less than 1.5% for the F4, F5, and F6 formulations.

The results indicate that, during storage, there was a slight shift to the right in the distribution, with increasing solid loading in the feed suspension and decreasing stabilizer amounts in the medium. However, a trend was observed for all suspensions: there was a small reduction in particle size and a shift to the smaller values in the PSD.

This was contrary to what was expected: the broadening of the PSD due to mild agglomeration that occurred during storage. This trend may be explained by stirring, but the suspensions were not kept under stirring conditions for the entire time of storage, only for the first three days following production. Another possible explanation could be the dynamic equilibrium between suspended ITZ particles in the media and soluble ITZ. Furthermore, the morphology, and thus the orientation, of the ITZ crystals could also have influenced the measured PSDs. As observed in previous SEM images, the crystals presented elongated plate shapes with irregular morphologies and were stacked on each other. However, since, in the laser diffraction technique, the particles are considered spherical, to determine the cause behind the decreasing trend in particle size, further, in-depth investigation is needed. In addition, no change in the solid-state form of the ITZ crystals (form I) was observed after 120 days of storage stability, as illustrated in Appendix A.

### 3.7. In Vitro Discriminatory Dissolution Comparison

In vitro dissolution testing was performed to discriminate between the optimized ITZ LAI formulations with a final solid loading of approximately 300 mg/g. These were compared against the suspension prepared with the as-received ITZ at an identical drug loading. The compositions of four ITZ suspension samples are described in Section 2.13. This comparative study assessed the effect of changing the process from MCR-CLASC to microfluidic-based SCT-CLASC along with downstream process intensification via membrane-based diafiltration (in F3 and F4) and the effect of reducing the stabilizer concentration on the final ITZ LAI suspensions (in F4 and F6). The relevant dissolution profiles are shown in Figure 11A,B.

As shown in Figure 11A,B, the raw API suspension dissolved slowly, with only 41% dissolved after 60 min, 60% dissolved after 240 min (4 h), and 100% dissolved after 4320 min (72 h). While more than 75% dissolution was accomplished after 60 min for all four produced suspensions, 100% dissolution was achieved after 2880 min (48 h). Due to the identical PSDs, morphology, and formulation compositions of the ITZ microparticles in the suspensions, no significant differences in the dissolution profiles of all three LAI suspensions were found. As illustrated in Figure 11B, the early phase of dissolution was slightly faster for F3|MCR-CLASC, followed by F4|SCT-CLASC. Due to a broader PSD and a lower stabilizer concentration compared to the other two suspensions, F6|SCT-CLASC had the slowest release profile amongst all three tested LAI suspensions. This result is explained by the fact that the particles had a higher surface area and, as a result, dissolved faster than the larger particles (as reported by API). The present study, which used ITZ as a model system, demonstrated that there was no difference in the IVR profiles between the two alternative CLASC procedures (MCR and SCT) followed by downstream processing studied when PSD, morphology, and formulation factors were kept as consistent as realistically feasible.

## 4. Conclusions

In the present work, after optimizing the operating parameters of the SCT-CLASC process, the most optimal suspension was generated by using a co-flow mixer, 20% *w*/*w* ITZ in NMP as the solvent, and 0.075% *w*/*w* Vit E TPGS 1000 aqueous solution as the antisolvent, with an S:AS ratio of 1:4, an S–AS flow rate of 5–20 mL/min, and an experimental and storage temperature of 25 °C. Therefore, the API-to-stabilizer ratio in the obtained suspension was low at 53:1. In addition, since the ITZ solid loading in the post-precipitation suspension was increased to up to 40 mg/g, it only had to be concentrated approximately 8 times to achieve the final LAI suspension with an ITZ solid loading of 300 mg/g by centrifugal filtration, compared to 30 times with the MCR-based CLASC setup. In comparison to the MCR-CLASC process, the SCT-CLASC method was able to produce suspensions with a higher solid loading while using a much lower amount of stabilizer, which is beneficial for a suspension’s safety profile, injectability, and sustainability. Further investigations need to be carried out to better understand the increased process efficiency in the microfluidic system. Additionally, both optimized post-precipitation suspensions and LAI suspensions showed controlled PSDs over time within the target range of 1–10 µm; however, a slight decreasing trend in particle size was observed during the stability assessment. The IVR study showed that there was no significant difference between changing production to the SCT-CLASC process and decreasing stabilizer concentration in the final formulation. As a result, this work demonstrates that the microfluidic approach can be a viable, energy-efficient, robust, and sustainable bottom-up alternative for the industrial-scale production of injectable suspensions of ITZ. Scale-up can be accomplished by parallelizing identical reactors, which is compatible with integration into a continuous manufacturing process. However, further investigations should be conducted on the upscaling towards an integrated, continuous manufacturing process for industrial implementation.

## Figures and Tables

**Figure 1 pharmaceutics-16-00376-f001:**
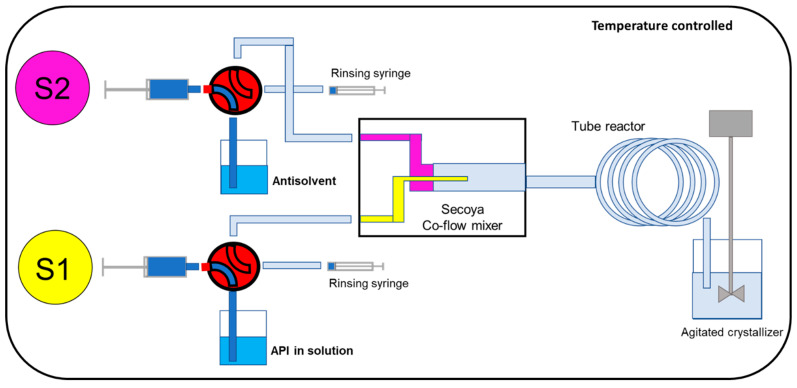
A schematic illustration of the continuous microfluidic antisolvent crystallization process. The solvent and antisolvent were both pumped into the mixing device, where crystallization of the drug particles occurred before they were collected in an agitated crystallizer [51].

**Figure 2 pharmaceutics-16-00376-f002:**
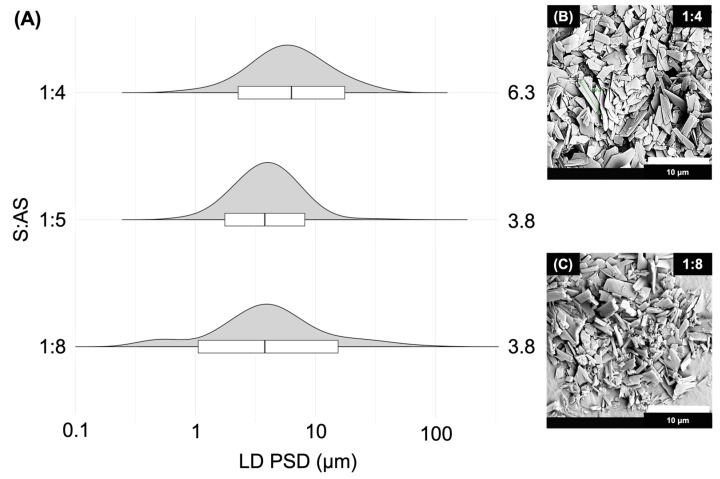
(**A**) Effect of solvent-to-antisolvent (S:AS) ratios on itraconazole (ITZ) particle size distribution (PSD), measured by laser diffraction (LD), in suspensions with solid loading of 10 mg/g (after 48 h of storage in an agitated crystallizer at 25 °C), generated via the Y mixer microchannel reactor-based continuous liquid antisolvent crystallization (MCR-CLASC) process. The violin plots orientated with horizontal density curves depict the whole PSDs. The results are presented in the overlaid box plots, where the medians correspond to the D50 values, the lower hinges correspond to the D10 values, the higher hinges correspond to the D90 values, and the whiskers correspond to the D1–D99 ranges. An increase in the S:AS ratio resulted in a gradual decrease in the median D50 value mentioned on the right and the length of the boxplot. Scanning electron microscopy (SEM) images of ITZ microparticles in the generated suspensions (after 48 h of storage in an agitated crystallizer at 25 °C) produced via Y mixer microchannel reactor-based continuous liquid antisolvent crystallization (MCR-CLASC) using S:AS ratios of (**B**) 1:4 and (**C**) 1:8, respectively.

**Figure 3 pharmaceutics-16-00376-f003:**
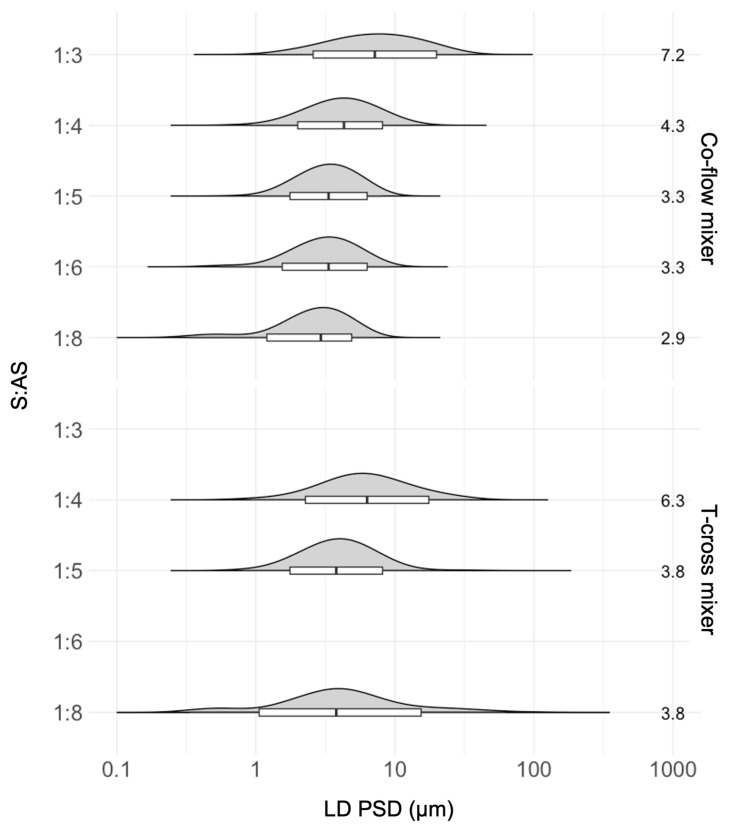
Effect of solvent-to-antisolvent (S:AS) ratios on the particle size distribution (PSD) of itraconazole (ITZ) in the generated suspensions (after 48 h of storage in an agitated crystallizer at 25 °C) via the Secoya microfluidic crystallization technology-based continuous liquid antisolvent crystallization (SCT-CLASC) process using comparisons between the co-flow mixer and the T-cross mixer, measured by laser diffraction (LD). The entire PSDs are shown in the violin plots that are orientated with horizontal density curves. The findings are displayed in the overlaid box plots, where the medians correspond to the D50 values, the lower hinges correspond to the D10 values, the higher hinges correspond to the D90 values, and the whiskers correspond to the D1–D99 ranges. The median D50 value for the S:AS ratio, shown on the right, decreases slightly as the ratio increases up to 1:4.

**Figure 4 pharmaceutics-16-00376-f004:**
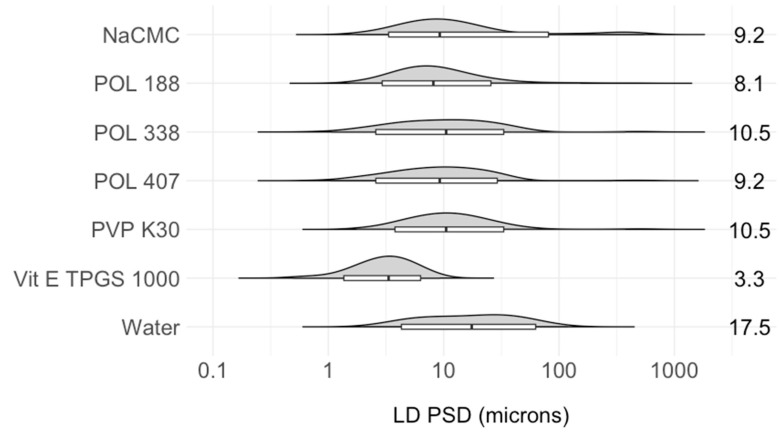
Impact of six water-soluble stabilizers (polymers and surfactants) on the ITZ particle size distribution (PSD) of the suspensions with a solid loading of 20 mg/g (after 48 h of storage at 25 °C in an agitated crystallizer) in comparison to reference water produced via the Secoya microfluidic crystallization technology-based continuous liquid antisolvent crystallization (SCT-CLASC) process using a co-flow mixer, measured by laser diffraction. The entire PSDs of ITZ are represented by the violin plots that are orientated with horizontal density curves. The overlaid box plots display the results, with the medians corresponding to the D50 values, the lower hinges corresponding to the D10 values, the higher hinges corresponding to the D90 values, and the whiskers corresponding to the D1–D99 ranges. The efficacy of the excipient in stabilizing the ITZ suspension increases with decreasing D50 values and box plot lengths, as indicated on the right.

**Figure 5 pharmaceutics-16-00376-f005:**
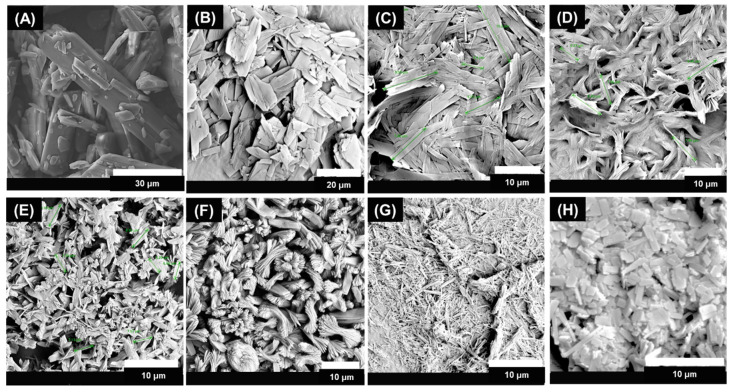
Scanning electron microscopy images of ITZ particles in suspensions with solid loading of 20 mg/g produced by the Secoya microfluidic crystallization technology-based continuous liquid antisolvent crystallization (SCT-CLASC) process using six water-soluble stabilizers in water, used as the antisolvent, compared to (**A**) ITZ as received and (**B**) reference water after 48 h of storage in an agitated crystallizer at 25 °C: (**C**) Poloxamer 188, (**D**) Poloxamer 338, (**E**) Poloxamer 407, (**F**) PVP K30, (**G**) Na CMC, and (**H**) Vitamin E TPGS. Green lines in the picture indicate measured particles size.

**Figure 6 pharmaceutics-16-00376-f006:**
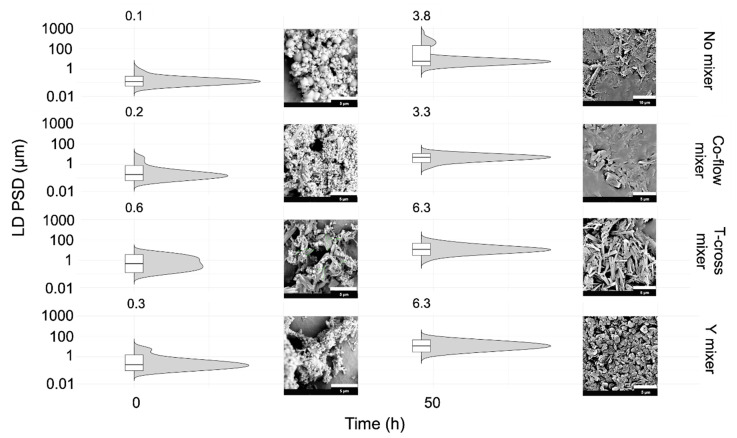
Comparison of the three microfluidic static mixers used (co-flow, T-cross and Y mixers) and the condition with no mixing device (no mixer) in terms of ITZ particle size distribution (PSD) in the generated suspensions with solid loading of 20 mg/g (just after generation and after 48 h of storage in an agitated crystallizer at 25 °C) via the Secoya microfluidic crystallization technology-based continuous liquid antisolvent crystallization (SCT-CLASC) process, measured by laser diffraction (LD). The entire PSDs are shown in the violin plots that are orientated with horizontal density curves. The superimposed box plots display the results, with the medians corresponding to the D50 values, the lower hinges corresponding to the D10 values, the higher hinges corresponding to the D90 values, and the whiskers corresponding to the D1–D99 ranges. The narrower the length of the box plot, the lower the D50 value mentioned at the top. Scanning electron microscopy images of initial ITZ particles in suspensions just after generation at 25 °C via the SCT-CLASC process and after 48 h of storage in an agitated crystallizer at 25 °C under the no mixer, co-flow mixer, T-cross mixer, and Y mixer conditions (top to bottom) beside the PSDs.

**Figure 7 pharmaceutics-16-00376-f007:**
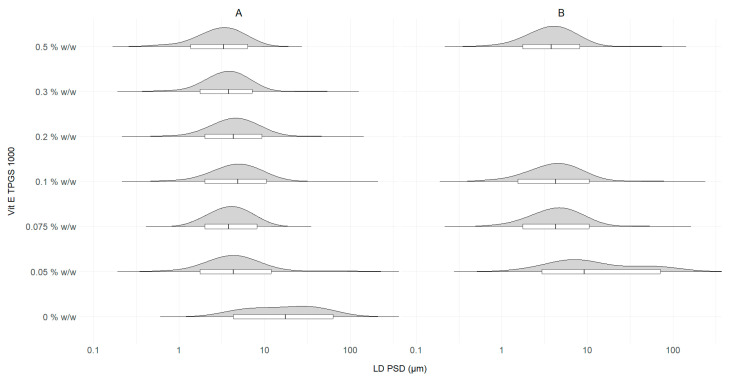
Effect of different Vit E TPGS 1000 concentrations in terms of ITZ particle size distribution (PSD) in the generated suspensions with solid loading of 20 mg/g (after 48 h of storage in an agitated crystallizer at 25 °C) via the Secoya microfluidic crystallization technology-based continuous liquid antisolvent crystallization (SCT-CLASC) process using a co-flow mixer (**A**) at the optimized S–AS flow rate of 5–20 mL/min and (**B**) at the highest S–AS flow rate of 10–40 mL/min, measured by laser diffraction (LD).

**Figure 8 pharmaceutics-16-00376-f008:**
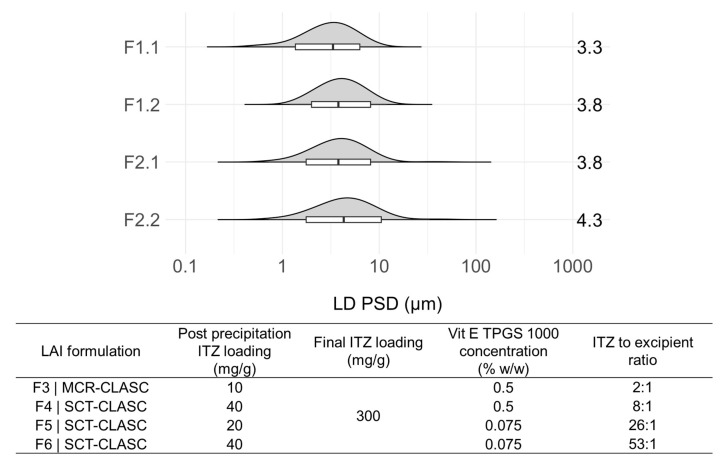
Comparison of the four optimized suspensions of itraconazole in terms of particle size distribution (PSD) generated via the Secoya microfluidic crystallization technology-based continuous liquid antisolvent crystallization (SCT-CLASC) process after 48 h of storage in an agitated crystallizer at 25 °C, measured by laser diffraction. The entire PSDs are represented by the violin plots that are orientated with horizontal density curves. The findings are displayed in the overlaid box plots, where the whiskers correspond to the D1–D99 ranges, the lower hinges correspond to the D10 values, the higher hinges correspond to the D90 values, and the medians correspond to the D50 values, indicated on the right.

**Figure 9 pharmaceutics-16-00376-f009:**
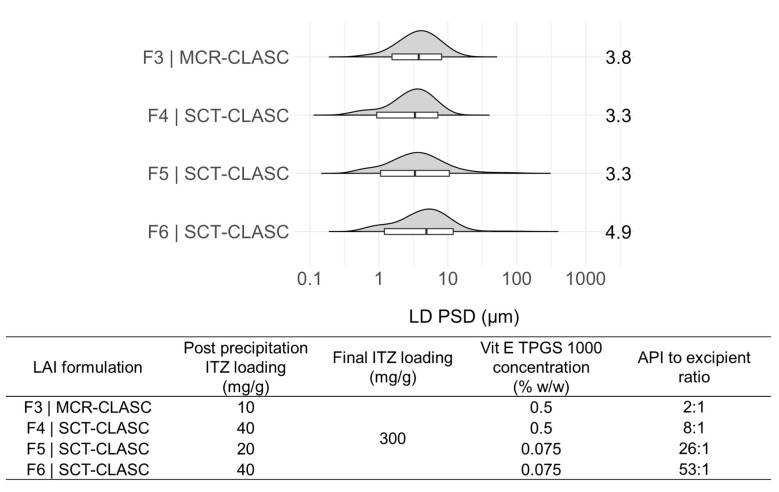
Comparison of four optimized LAI suspensions of itraconazole with a final solid loading of 300 mg/g in terms of particle size distribution (PSD) generated via an integrated bottom-up process (solvent switch using membrane diafiltration after the Secoya microfluidic crystallization technology-based continuous liquid antisolvent crystallization (SCT-CLASC) process), measured by laser diffraction (LD). The violin plots orientated with horizontal density curves depict the whole PSDs. The results are presented in the overlaid box plots, where the medians correspond to the D50 values mentioned on the right, the lower hinges correspond to the D10 values, the higher hinges correspond to the D90 values, and the whiskers correspond to the D1–D99 ranges.

**Figure 10 pharmaceutics-16-00376-f010:**
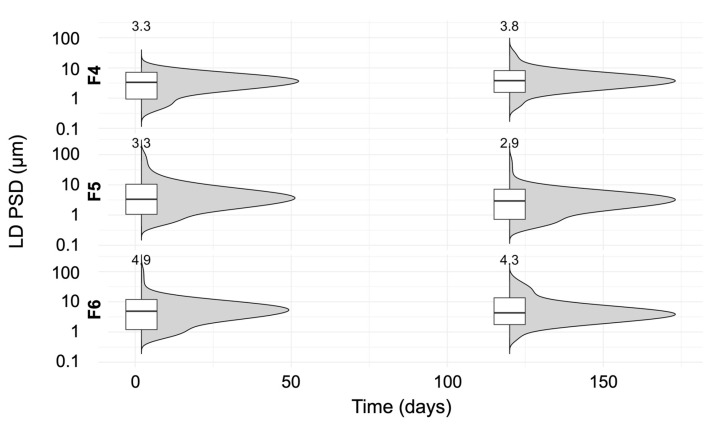
Particle size distribution (PSD) comparison of three optimized LAI microsuspensions of itraconazole (with a final solid loading of 300 mg/g) in terms of particle size distribution (PSD) generated via downstream processing after the Secoya microfluidic crystallization technology-based continuous liquid antisolvent crystallization (SCT-CLASC) process, analyzed by laser diffraction after 7 and 120 days of storage at 25 °C. The entire PSDs are shown in the violin plots that are arranged with vertical density curves. The findings are displayed in overlaid box plots, in which the whiskers correspond to the D1–D99 ranges, the lower hinges correspond to the D10 values, the higher hinges correspond to the D90 values, and the medians, which are indicated on top of each violin plot, correspond to the D50 values. F4: post-precipitation feed suspension solid loading of 40 mg/g, Vit E TPGS 1000 0.5% *w*/*w*; F5: post-precipitation feed suspension solid loading of 20 mg/g, Vit E TPGS 1000 0.075% *w*/*w*; F6: post-precipitation feed suspension solid loading of 40 mg/g, Vit E TPGS 1000 0.075% *w*/*w*.

**Figure 11 pharmaceutics-16-00376-f011:**
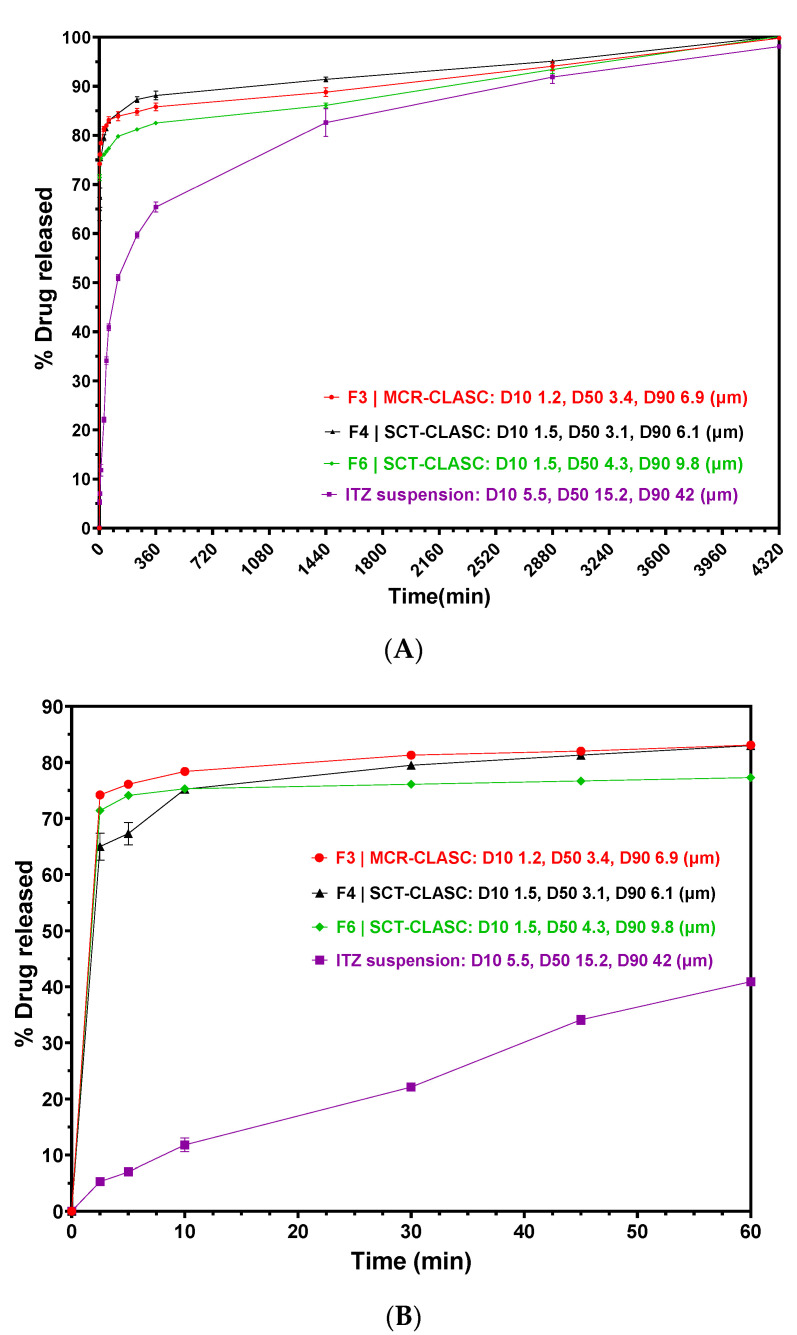
Comparison of in vitro release profiles of itraconazole (ITZ) LAI microsuspensions (with a final solid loading of 300 mg/g) expressed as a percentage of drug released over time: (**A**) after 4320 min (72 h); (**B**) an enlarged view of initial timepoints up to 60 min. F3/MCR-CLASC: ITZ suspension with post-precipitation solid loading of 10 mg/g, Vit E TPGS 1000 0.5% *w*/*w*, produced by the microchannel reactor-based continuous liquid antisolvent crystallization process; F4/SCT-CLASC: ITZ suspension with post-precipitation solid loading of 40 mg/g, Vit E TPGS 1000 0.5% *w*/*w*, produced by the Secoya microfluidic crystallization technology-based continuous liquid antisolvent crystallization (SCT-CLASC) process; F6/SCT-CLASC: ITZ suspension with post-precipitation solid loading of 40 mg/g, Vit E TPGS 1000 0.075% *w*/*w*, produced by the SCT-CLASC process; ITZ suspension: as-received ITZ API suspension in 0.5% *w*/*w* Vit E TPGS 1000 medium. Error bars correspond to standard deviations for *n* = 6 for each group.

**Table 1 pharmaceutics-16-00376-t001:** Equilibrium solubility of itraconazole in six stabilizer solutions at 25 °C (*n* = 3).

Type of Stabilizer	Concentration (% *w*/*w*)	Equilibrium Solubility (μg/g Solution)
Vit E TPGS 1000	0.5	12 ± 1.2
POL 188	0.5	4.4 ± 0.3
POL 338	0.5	8 ± 0.5
POL 407	0.5	9.5 ± 0.5
PVP K30	0.5	3.3 ± 0.2
Na CMC	0.5	2 ± 0.1
Vit E TPGS 1000	0.5	12 ± 1.2
	0.075	0.2 ± 0.05

**Table 2 pharmaceutics-16-00376-t002:** Overview of final ITZ solid loading, Vit E TPGS 1000, and residual organic solvent concentrations in the LAI suspensions after downstream processing (solvent switch using membrane diafiltration and reconstitution).

LAI Formulation	Final ITZ Loading (% *w*/*w*)	Vit E TPGS 1000 (% *w*/*w*)	Residual NMP (% *w*/*w*)
F3|MCR-CLASC	29 ± 1.8	14.4 ± 0.6	<0.05
F4|SCT-CLASC	30 ± 2.1	3.7 ± 0.1	
F5|SCT-CLASC	29 ± 2.4	1.1 ± 0.1	
F6|SCT-CLASC	30 ± 1.6	0.5 ± 0.1	

## Data Availability

The data presented in this study are available in this article and Appendix A.

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
