# Peer review of "Continuous Microfluidic Antisolvent Crystallization as a Bottom-Up Solution for the Development of Long-Acting Injectable Formulations"

_pharmaceutics, 2024, doi:10.3390/pharmaceutics16030376_

Round 1
Reviewer 1 Report
Comments and Suggestions for Authors
Review of manuscript “Continuous microfluidic antisolvent crystallization as a bottom-up solution for the development of long-acting injectable formulations” by Nandi et al.
The authors applied Secoya microfluidic crystallization technology based continuous liquid antisolvent crystallization (SCT-CLASC) process to produce itraconazole (ITZ) long-acting injectable (LAI) suspension. First they optimized process parameters and then monitored the quality of produced suspensions. The authors, have assessed suitability of SCT-CLASC to the industrial-scale production of injectable suspensions.
Major concerns:
1) Methods
In my opinion, when sending an article for review, authors should avoid the following statements: "as described in the literature (accepted for publication)", because currently there is no publication to which they could refer. Of course, I do not suspect any ill will on the part of the authors, but only the lack of certain materials (those publications) that would facilitate the review of the article.
Therefore, the authors, although they refer to publications that will be published in the future, should include at least a minimum of information and/or figures/diagrams that will allow readers to learn about the methods used.
Please discuss more on apparatus (line 387) and MCR-CLASC method (line 161)
2) Methods and Results
The reader may get a bit lost when looking at the results of 'in vitro' tests; the authors are developing an LAI formulation, i.e. sustained release, but in the figure (and in the experimental setup) it can be seen that the release is almost immediate (in the sense of EMA/FDA guidelines, > 80% in 30 min). Therefore, do 'in vitro' tests translate into 'prolongation' of the release of the medicinal substance?
Could the authors discuss what is the rationale behind this particular ‘in vitro’ experimental method?
3) Results
The stabilizers used in the research also have a solubilizing effect, has this fact been taken into account? Was it tested if it is influencing crystallization for example, by testing the solubility of itraconazole in stabilizer solutions (in particular as Vit E TPGS 1000, Poloxamers, NaCMC, PVPK30)
4) Results
Stability results would need to be statistically analyzed to indicate that differences may actually result in changes in drug formulation. In my opinion, tailing (D10 fraction) in F4, F5, F6 formulations may cause some problems (in the future), while in F3 tailing is not observed; this D10 faction will be the most susceptible to change;
Minor concerns:
1) Line 163 – there is no Figure 3.2. Please correct.
Comments on the Quality of English LanguageMinor editing of English language required, can be done after acceptance of the manuscript.
Author Response
We thank the reviewer for these observations and suggestions.
Major concerns:
1) Methods
In my opinion, when sending an article for review, authors should avoid the following statements: "as described in the literature (accepted for publication)", because currently there is no publication to which they could refer. Of course, I do not suspect any ill will on the part of the authors, but only the lack of certain materials (those publications) that would facilitate the review of the article. Therefore, the authors, although they refer to publications that will be published in the future, should include at least a minimum of information and/or figures/diagrams that will allow readers to learn about the methods used.
Manuscript page number 7, section 2.2, and line number 214
In the revised manuscript, we removed the statements and added reference to our earlier published work. The following text has been added:
“as explained in detail in the earlier work published by our group [54]”
Please discuss more on apparatus (line 387) and MCR-CLASC method (line 161)
Manuscript page number 14, section 2.13, and line number 450
In the revised manuscript, we referred to our earlier published work. The following text has been added:
“as described in the earlier work published by our group [54]”
2) Methods and Results
The reader may get a bit lost when looking at the results of 'in vitro' tests; the authors are developing an LAI formulation, i.e. sustained release, but in the figure (and in the experimental setup) it can be seen that the release is almost immediate (in the sense of EMA/FDA guidelines, > 80% in 30 min). Therefore, do 'in vitro' tests translate into 'prolongation' of the release of the medicinal substance?
Could the authors discuss what is the rationale behind this particular ‘in vitro’ experimental method?
Manuscript page number 14, section 2.13, and line number 434
We updated the section title to “in vitro discriminatory dissolution comparison” to avoid further confusion.
We thank the reviewer for their observations and suggestions. In the revised manuscript, we clearly stated the rationale for conducting in vitro discriminatory dissolution testing to understand any difference in ITZ LAI microsuspension formulation release profiles due to changes in process and stabilizer concentration. This discriminatory dissolution testing is not performed to provide any evidence towards prolonged release behavior from the ITZ suspensions; it is only to assess the quality of the generated injectable suspensions and evaluate whether the newly developed integrated SCT-CLASC process can effectively facilitate the production of ITZ LAI suspensions.
Manuscript page numbers 31-32, section 3.7, and line numbers 942 - 960
The following text has been added to the revised manuscript:
“In vitro dissolution testing was performed to discriminate among the release profiles of different optimized ITZ LAI formulations with a final solid loading of approximately 300 mg/g. when compared against the as-received ITZ suspension of identical loading. The findings of comparative in vitro discriminatory dissolution profiles of optimized ITZ LAI microsuspensions produced by the SCT-CLASC process (F4 | SCT-CLASC, F5 | SCT-CLASC), and the MCR-CLASC process (F3 | MCR-CLASC) are shown in Figure 11 (A) and (B). The composition of four ITZ suspension samples is described in Section 2.13. To assess the effect of changing the process from MCR-CLASC to microfluidic-based SCT-CLASC along with downstream process intensification via membrane-based diafiltration (between F3 and F4) and the effect of reducing stabilizer concentration on the final ITZ LAI suspensions (between F4 and F6), the in vitro release profiles were compared in discriminatory dissolution media to reveal any differences. After conducting a dissolution screening study of ITZ, which is beyond the scope of this work, 1% w/v Tween 20 with 0.1N HCl of pH 1.2 in 500 mL of media was selected as the discriminatory medium for testing ITZ LAI suspensions based on particle size and achieving complete release with a standard profile. To maintain sink condition while dosing a reasonable amount of formulation of practically water insoluble ITZ in in vitro release testing, a highly acidic pH of 1.2 and a large volume of dissolution medium (500 mL) had to be chosen.”
3) Results
The stabilizers used in the research also have a solubilizing effect, has this fact been taken into account? Was it tested if it is influencing crystallization for example, by testing the solubility of itraconazole in stabilizer solutions (in particular as Vit E TPGS 1000, Poloxamers, NaCMC, PVPK30)
In the revised manuscript, we have incorporated the solubility study of ITZ in stabilizers’ solution and reported experimentally determined equilibrium solubilities of ITZ in 0.5 % w/w aqueous solution of stabilizers (Vit E TPGS 1000, POL 188, POL 338, POL 407, NaCMC, and PVPK30) and 0.075 % w/w Vit E TPGS solution. Itraconazole is a water insoluble compound with the reported solubility value of 1 ng / g solution in water [54]. Although the addition of stabilizers increases the solubility of ITZ up to a few µg / g solution levels, the values are still in the very poorly soluble range. This has a negligible impact on the crystallization process and yield where the concentration of ITZ feed solution is several orders of magnitude higher (in the range of 100–200 mg/g solution).
Manuscript page number 9, section 2.4.3 of the Method, and line numbers 281 – 285
The following text has been added to the revised manuscript:
“An excess amount of ITZ was added to each 10 g of 0.5% w/w aqueous stabilizer solutions, and the slurries were equlibriated for 72 h on stirring at 800 rpm at 25 °C. The ITZ concentrations were determined by chromatography analysis, as reported in the earlier work published by our group [54]. All solubility experiments were conducted in triplicate, and the standard deviations were reported accordingly.”
Manuscript page number 19, section 3.2.1.2 in Results and Discussion, and line numbers 581 – 591
The following text has been added to the revised manuscript:
“The results of the equilibrium solubility of ITZ in these six stabilizer solutions are shown in Table 1. It can be observed that the solubility increases significantly from ng/g solution to µg/g solution level when compared against pure water [54], probably due to the solubilization effect of the stabilizers. The equilibrium solubility for the six investigated surfactants varies between 2-13 µg/g solution at 25°C, which is still in the very poorly soluble range, while in the case of the antisolvent crystallization process, the ITZ feed solution concentration is several folds higher at 100 mg / g solution. The highest solubility for ITZ is obtained in Vit E TPGS 1000, followed by POL 407 at a similar concentration. ITZ solubility in an aqueous solution of Vit E TPGS 1000 decreases in the presence of a lower stabilizer concentration in the media.”
Table 1: Equilibrium solubility of itraconazole in six stabilizer solutions at 25°C (n=3).
|
Type of stabilizer |
Concentration (% w/w) |
Equilibrium solubility (μg/g solution) |
|
Vit E TPGS 1000 |
0.5 |
12 ± 1.2 |
|
POL 188 |
0.5 |
4.4 ± 0.3 |
|
POL 338 |
0.5 |
8 ± 0.5 |
|
POL 407 |
0.5 |
9.5 ± 0.5 |
|
PVP K30 |
0.5 |
3.3 ± 0.2 |
|
Na CMC |
0.5 |
2 ± 0.1 |
|
Vit E TPGS 1000 |
0.5 |
12 ± 1.2 |
|
0.075 |
0.2 ± 0.05 |
4) Results
Stability results would need to be statistically analyzed to indicate that differences may actually result in changes in drug formulation. In my opinion, tailing (D10 fraction) in F4, F5, F6 formulations may cause some problems (in the future), while in F3 tailing is not observed; this D10 faction will be the most susceptible to change;
Manuscript page number 30, section 3.6, and line numbers 911 - 934
We thank the reviewer for this observation and suggestion. In the revised manuscript, we included the range of D10, D50, and D90 values after 120 days of stability with a standard deviation (n = 3). Also, the % relative standard deviation of the D10 values of F4, F5, and F6 formulations after 120 days of storage was mentioned.
In Figure 10, the PSDs are displayed in an overlaid box plot, in which the whiskers extend from D1 to D99 along with the violin plot. The lower hinge of the box plot corresponds to D10, the higher hinge to D90, and the median, which is also indicated on top of each violin plot, corresponds to D50. Through our paper, we tried to highlight the importance of showing the full size and shape of distribution instead of only describing the D50 and span values which can often be misinterpreted. When compared, the D10 value, which is the lower hinge of the boxplot, does not present a significant change for all three tested formulations after 7 and 120 days of stability at 25 °C. The probable reason for the lower shift in tailing, i.e., the D1 and D99 values of the violin plot, which the reviewer pointed out, is explained in the original submitted manuscript.
The following text has been updated to the revised manuscript:
“The D10 values differed marginally between 0.8 ± 0.1 to 1.6 ± 0.2 µm; the D50 values varied slightly between 2.9 ± 0.4 to 4.3 ± 0.3 µm, whereas the D90 values changed between 9.1 ± 0.7 to 11.2 ± 0.5 µm for the three LAI suspensions. The relative standard deviation (RSD) for the D10 values after 120 days of storage is less than 1.5 % for all F4, F5, and F6 formulations. These results indicate that during storage, there was a slight shift to the right in distribution with increasing solid loading in feed suspension and decreasing stabilizer amount in the media. This was contrary to what was expected: a broadening of the PSD due to mild agglomeration that occurred during storage. This trend may be explained by stirring, but the suspensions were not kept under stirring conditions for the entire time of storage, only for the first three days following production. Another possible explanation could be the dynamic equilibrium between suspended ITZ particles in the media and soluble ITZ.”
Minor concerns:
1) Line 163 – there is no Figure 3.2. Please correct.
Manuscript page number 7, section 2.2, and line numbers 215-216
In the revised manuscript, we corrected the line: “(Y-mixer as fluidic mixing device)”.
Reviewer 2 Report
Comments and Suggestions for Authors
The manuscript is well written. I would like the authors to address the following
Abstract - Line 20 and 21 replace the word solution with suspension
Please report the solubility data of ITZ in various stabilizer solutions at their respective concentration
The in-vitro release study indicates that 80% of the drug is dissolved within the first 60 min, how do the authors justify the intended use of the mirosuspension as a long acting product??
Comments on the Quality of English LanguageNA
Author Response
We thank the reviewer for these observations and suggestions.
The manuscript is well written. I would like the authors to address the following
Abstract - Line 20 and 21 replace the word solution with suspension
Manuscript page number 2, section Abstract, and line number 43, 44, 46
We updated accordingly in the revised manuscript.
Please report the solubility data of ITZ in various stabilizer solutions at their respective concentration
Manuscript page number 9, section 2.4.3 of method section, and line numbers 281 – 285 and
manuscript page number 19, section 3.2.1.2 in Results and Discussion, and line numbers 580 – 591
As per reviewer suggestion, we reported the solubility data of itraconazole in all tested stabilizer solutions at their respective concentrations. This comment has been addressed before in the Reviewer 1 comment section.
Table 1: Equilibrium solubility of itraconazole in six stabilizer solutions at 25°C (n=3).
|
Type of stabilizer |
Concentration (% w/w) |
Equilibrium solubility (μg/g solution) |
|
Vit E TPGS 1000 |
0.5 |
12 ± 1.2 |
|
POL 188 |
0.5 |
4.4 ± 0.3 |
|
POL 338 |
0.5 |
8 ± 0.5 |
|
POL 407 |
0.5 |
9.5 ± 0.5 |
|
PVP K30 |
0.5 |
3.3 ± 0.2 |
|
Na CMC |
0.5 |
2 ± 0.1 |
|
Vit E TPGS 1000 |
0.5 |
12 ± 1.2 |
|
0.075 |
0.2 ± 0.05 |
The in-vitro release study indicates that 80% of the drug is dissolved within the first 60 min, how do the authors justify the intended use of the microsuspension as a long acting product??
Manuscript page numbers 31-32, section 3.7, and line numbers 942 – 960
A similar comment was addressed earlier in the Reviewer 1 and Reviewer 2 comments sections.
In vitro discriminatory dissolution studies were carried out with the objective to evaluate and understand any difference in formulation release pattern due to a change in manufacturing process from MCR-CLASC to SCT-CLASC process (F3 and F4) keeping all other variables constant, and also due to a 26-fold reduction in Vit E TPGS 1000 concentration in final formulations (between F4 (0.5% w/w) and F6 (0.075% w/w)).
This discriminatory dissolution testing is not performed to provide any evidence towards prolonged release behavior from the ITZ microsuspensions, it is only to assess the quality of the generated injectable microsuspensions and evaluate whether the newly developed integrated SCT-CLASC process can effectively facilitate the production of ITZ LAI suspensions. A proof-of-concept study of physiologically relevant biomimetic dissolution studies in various setups for the LAI suspensions was reported in the earlier work published by our group [54].
Reviewer 3 Report
Comments and Suggestions for Authors
The article titled: "Continuous Microfluidic Antisolvent Crystallization: A Bottom-Up Solution for the Development of Long-Acting Injectable Formulations" presents a compelling study on the application of Secoya microfluidic crystallization technology (SCT-CLASC) for producing long-acting injectable (LAI) microsuspensions, using itraconazole (ITZ) as a model drug. The research outlines a bottom-up approach aimed at overcoming the limitations of traditional top-down manufacturing methods. By optimizing the SCT-CLASC process and comparing it with a previously developed microchannel reactor-based method (MCR-CLASC), the authors successfully demonstrate the production of stable LAI formulations with high drug concentration and efficient use of excipients.
There are plenty of details related to the experimental setup and provided results which is a very strong point of the presented work. There is just one concern I would like to discuss more in-depth - the choice of a highly acidic pH (1.2) and a large volume (500 mL) for dissolution testing. It seems incongruent with the physiological conditions typical of subcutaneous (SC) or intramuscular (IM) environments, where LAI formulations are intended for use. Did authors try other dissolution setups (reflecting application site) to compare obtained products? If not why? If yes - why these particular results were picked for presentation?
Author Response
We thank the reviewer for these observations and suggestions.
There are plenty of details related to the experimental setup and provided results which is a very strong point of the presented work. There is just one concern I would like to discuss more in-depth - the choice of a highly acidic pH (1.2) and a large volume (500 mL) for dissolution testing. It seems incongruent with the physiological conditions typical of subcutaneous (SC) or intramuscular (IM) environments, where LAI formulations are intended for use. Did authors try other dissolution setups (reflecting application site) to compare obtained products? If not why? If yes - why these particular results were picked for presentation?
Manuscript page numbers 31-32, section 3.7, and line numbers 942 – 960
In vitro discriminatory dissolution studies were carried out with the objective to evaluate and understand any difference in formulation release pattern due to a change in manufacturing process from MCR-CLASC to SCT-CLASC process (F3 and F4) keeping all other variables constant, and also due to a 26-fold reduction in Vit E TPGS 1000 concentration in final formulations (between F4 (0.5% w/w) and F6 (0.075% w/w)).
Based on the findings of a dissolution media screening study for ITZ (which is beyond the scope of this work, Janssen Pharmaceutica internal knowledge), 500 mL of 1% w/v Tween 20 with 0.1N HCl at pH 1.2 was chosen as the discriminatory medium in order to test ITZ LAI suspensions according to particle size discrimination and achieving complete release (100%) with a standard profile. An extremely acidic pH of 1.2 and a considerable volume of dissolving media (500 mL) had to be used in order to sustain sink conditions while dosing an acceptable amount of practically water insoluble API, ITZ, for in vitro release studies.
This study was not performed to establish any biomimetic prolonged release behavior from ITZ LAI suspension. A proof-of-concept study of physiologically relevant biomimetic dissolution studies in various setups was reported in the earlier work published by our group [54].
The following text has been added to the revised manuscript, as also mentioned before while addressing reviewer 1’s similar comment:
“In vitro dissolution testing was performed to discriminate among the release profiles of different optimized ITZ LAI formulations with a final solid loading of approximately 300 mg/g. against the as-received ITZ suspension of identical loading. The findings of comparative in vitro discriminatory dissolution profiles of optimized ITZ LAI microsuspensions produced by the SCT-CLASC process (F4 | SCT-CLASC, F5 | SCT-CLASC), and the MCR-CLASC process (F3 | MCR-CLASC) are shown in Figure 11 (A) and (B). The composition of four ITZ suspension samples is described in Section 2.13. To assess the effect of changing the process from MCR-CLASC to microfluidic-based SCT-CLASC along with downstream process intensification via membrane-based diafiltration (between F3 and F4) and the effect of reducing stabilizer concentration on the final ITZ LAI suspensions (between F4 and F6), the in vitro release profiles were compared in discriminatory dissolution media to reveal any differences. After conducting a dissolution screening study of ITZ, which is beyond the scope of this work, 1% w/v Tween 20 with 0.1N HCl of pH 1.2 in 500 mL of media was selected as the discriminatory medium for testing ITZ LAI suspensions based on particle size and achieving complete release with a standard profile. To maintain sink condition while dosing a reasonable amount of formulation of practically water insoluble ITZ in in vitro release testing, a highly acidic pH of 1.2 and a large volume of dissolution medium (500 mL) had to be chosen.”
Round 2
Reviewer 1 Report
Comments and Suggestions for Authors
I would like to thank the authors for their work. All points raised were addressed, therefore I recommend the article to be further processed.
Comments on the Quality of English LanguageMinor editing of English language required, which could be done by Editorial Office.
Reviewer 2 Report
Comments and Suggestions for Authors
NA